



# Vortex streets to the lee of Madeira in a km-resolution regional climate model

Qinggang Gao[1, a, b], Christian Zeman[1], Jesus Vergara-Temprado[1, c], Daniela C.A. Lima[3], Peter Molnar[2], and Christoph Schär[1]

[1]Institute for Atmospheric and Climate Science, ETH Zürich, Zürich, Switzerland
[2]Institute of Environmental Engineering, ETH Zürich, Zürich, Switzerland
[3]Instituto Dom Luiz (IDL), Faculdade de Ciências, Universidade de Lisboa, 1749-016 Lisboa, Portugal
[a]Now at Ice Dynamics and Paleoclimate, British Antarctic Survey, Cambridge, United Kingdom
[b]Now at Department of Earth Sciences, University of Cambridge, Cambridge, United Kingdom
[c]Now at UBS, Zurich, Switzerland

**Correspondence:** Qinggang Gao (qino@bas.ac.uk)

**Abstract.** Atmospheric vortex streets are a widely studied dynamical effect of isolated mountainous islands. Observational evidence comes from case studies and satellite imagery, but the climatology and annual cycle of vortex shedding are often poorly understood. Using the non-hydrostatic limited-area COSMO model driven by the ERA-Interim reanalysis, we conducted a ten-year-long simulation over a mesoscale domain covering Madeira and Canary Archipelagos at high spatial (grid spacing 1 km) and temporal resolutions. Basic properties of vortex streets were analyzed and validated through a 6-day-long case study in the lee of Madeira Island. The simulation compares well with satellite and aerial observations and with existing literature on idealized simulations. Our results show a strong dependency of vortex shedding on local and synoptic flow conditions, which are to a large extent governed by the location, shape, and strength of the Azores high. As part of the case study, we developed a vortex identification algorithm. The algorithm is based on a set of objective criteria and enabled us to develop a climatology of vortex shedding from Madeira Island for the 10-year simulation period. The analysis shows a pronounced annual cycle with an increasing vortex shedding rate from April to August and a sudden decrease in September. This cycle is consistent with mesoscale wind conditions and local inversion height patterns.

## 1 Introduction

Flow around isolated mountainous islands can lead to the formation of atmospheric vortex streets. Favorable conditions for the formation of vortex streets are quasi-steady upstream winds and a well-mixed planetary boundary layer capped by a thermal inversion below the island top (Etling, 1989).

Despite a striking similarity with the classic Kármán vortex street (Kármán, 2013), atmospheric vortex streets have different vorticity generation and decaying mechanisms. While Kármán vortex streets result from the instability of the wake behind cylinders in uniform laboratory flows, atmospheric vortex streets form in a stably-stratified turbulent flow passing asymmetric flat obstacles. The wake vorticity of Kármán vortex streets is generated by a viscous boundary layer, whereas vortex shedding can also be observed with free-slip boundary conditions in 3D numerical simulations. Atmospheric vortex shedding is specu-



lated to be a result of potential vorticity creation (Etling, 1989) and tilting of baroclinically generated horizontal vorticity under the deformation of isentropes that can be predicted by linear gravity wave theory (Smolarkiewicz and Rotunno, 1989). Wake patterns and vertical variations of Kármán vortex streets in unstratified fluids and a nonrotating reference frame depend solely

on the Reynolds number $Re$. In contrast, atmospheric vortex streets are also characterized by the Froude number $Fr$, Rossby number $Ro$, and Strouhal number $St$ (Horváth et al., 2020), whose definitions are given in Section 2.3.

Many studies focused on one or several specific aspects of vortex streets, which can be roughly divided into three categories: geometry, kinematics, and dynamics.

Geometry describes the shape of individual vortices and vortex streets. The total length of vortex streets can reach 400 km

or more (Etling, 1989), which indicates a lifetime of over 10 hours considering a mean flow advection velocity of $10\ \mathrm{m\,s^{-1}}$. Based on satellite images from the Aqua and Terra Moderate Resolution Imaging Spectroradiometer (MODIS) satellites, Young and Zawislak (2006) investigated aspect ratios of vortex streets, i.e. the ratio of the distance between two vortex tracks to the distance between neighboring like-rotating vortices. They reported higher ratios (0.36 - 0.47) compared to the value 0.28 derived by Kármán (2013) for Kármán vortex streets.

Kinematics of vortex streets include the shedding frequency and advection velocity of vortices. Using the Weather Research and Forecasting (WRF) model, Nunalee and Basu (2014) performed a detailed analysis of the Strouhal number. The Strouhal number represents a direct relation between vortex shedding frequency, wind velocity, and the crosswind island diameter at the inversion base. Tsuchiya (1969), using satellite images in the wake of Cheju Island, South Korea, reported that the advection velocity of vortices approximates 76% of the undisturbed flow when the aspect ratio is 0.332.

Dynamics of vortex streets refer to vortex generation, shedding, and decaying mechanisms. Heinze et al. (2012) employed an idealized large eddy simulation to study the azimuthally averaged structure of vortices. They found a warm core with convergence at the surface and divergence at the vortex top, and suggested that a sinking inversion at the vortex center might explain cloud-free eyes in observations. From a mathematical perspective, the vertical vorticity can be derived from the tilting of horizontal vorticity and turbulent stresses (Epifanio, 2015), whereas Rotunno et al. (1999) argued that the latter is inessential.

Schär and Smith (1993a) performed numerical simulations of a single layer of symmetric shallow-water flow past a circular mountain in a nonrotating environment with frictionless surface. They proposed two mechanisms for the production of potential vorticity. First, internal dissipation, which is derived from a theory indicating that the steady-state Bernoulli function is the stream function of the total vorticity flux. Second, flow separation leads to the joining of two flow streams with different values of the Bernoulli function. Subsequently, Schär and Smith (1993b) relaxed the symmetric flow conditions and analyzed

observed vortex streets with instability theory. With continuously stratified flow models, Schär and Durran (1997) used the dimensionless mountain height to identify different flow conditions that are conducive for vortex shedding and gravity wave breaking, respectively. They suggested the generalized Bernoulli theorem (Schär, 1993) for the explanation of potential vorticity generation. The dynamics of shedding in geophysical wakes are very similar to those of homogenous wakes, which have been well documented (Epifanio, 2015).

However, the study of vortex shedding in the lee of isolated islands is still limited by the availability of wind fields with high spatial and temporal resolutions. Satellite images of stratocumulus clouds, which are capped by an inversion base and can



act as tracers of vortex streets, have been widely used for the study of vortex shedding (Young and Zawislak, 2006; Etling, 2019). They can provide information about geometric characteristics of vortex streets through daily snapshots in the case of polar-orbiting satellites such as MODIS Terra and Aqua, but they are restricted to cloudy conditions (Young and Zawislak, 2006). With cloud motion winds and ocean surface winds from the Advanced Scatterometer (ASCAT), Horváth et al. (2020) reported an asymmetric vortex decay with larger peak vorticity in cyclonic eddies than in anticyclonic eddies at Guadalupe island. They argued that it is ascribed to Guadalupe's nonaxisymmetric shape and the Earth's rotation, where the latter factor is also supported by experiments of Ruppert-Felsot et al. (2005). Ito and Niino (2015) adopted a non-hydrostatic mesoscale model for numerical simulations of vortex streets in the lee of Cheju Island. They speculated that vortices are more likely generated through gravity wave breaking and the baroclinicity of flows over mountains than through viscous boundary layers. The importance of air-sea interactions on the wake patterns of Madeira Island was studied using the WRF numerical model by Caldeira and Tomé (2013). They proposed that the enhanced vertical mixing by increased sea surface temperature (SST) could lead to the erosion of the inversion base and a transition from vortex-shedding to a long straight wake. Aerial observations were employed for the study of atmospheric wakes in the lee of Hawaii (Smith and Grubišić, 1993) and Madeira Island (Grubišić et al., 2015). Grubišić et al. (2015) recommended potential vorticity, which can be generated by gravity wave breaking and is conservative in an adiabatic and inviscid flow, for the study of vorticity sources rather than the nonconservative vertical vorticity.

The impacts of vortex streets on the atmosphere and ocean have received relatively little attention. Using the Regional Oceanic Modeling System ocean circulation model, Couvelard et al. (2012) investigated the influence of wakes induced by Madeira Island on the ocean-eddy generation with an eddy tracking algorithm developed by Nencioli et al. (2010). They proposed that the island-induced wind-wake contributes to oceanic surface kinetic energy and vorticity, as well as the generation and containment of ocean eddies. Nunalee et al. (2015) used a coupled mesoscale model and ray-tracing framework to explore km-scale optical refractive perturbations caused by wake vortices of three islands. These perturbations are crucial for long-range optical communication systems that depend on precise laser guidance. Their study highlights the necessity of considering heterogeneous atmospheres (especially in the presence of vortex streets) for optical wave propagation studies. With a focus on atmosphere-ocean interactions, Azevedo et al. (2020) studied the detailed responses of upper oceanic profiles of temperature, salinity, and turbulence to the Madeira warm wakes, which mainly result from intense solar radiation in cloud-free conditions.

In this study, we focused on vortex streets in the lee of Madeira Island. It involves not only a case study as in most previous studies but also a climatology over one decade. The Madeira Archipelago is located near the northwestern African coast and forms part of Macaronesia. The weather and climate patterns in Macaronesia are mainly influenced by the subtropical semi-permanent Azores high-pressure system (Davis et al., 1997), prevailing north-easterly trade winds (Cropper and Hanna, 2014), local orographic characteristics (Carrillo et al., 2016), surrounding ocean currents, and Sahara dust advection (Cropper, 2013). Madeira Island is the largest ($740 \text{ km}^2$) and highest (1862 m) island in this archipelago and the major source of atmospheric flow disturbances, while Porto Santo and Desertas play a minor role due to low altitudes. Madeira Island experiences less precipitation in summer than in winter. This can be mostly attributed to the more northern position of the Intertropical Convergence Zone (ITCZ) in summer and its associated dry region from the Hadley circulation. More precipitation is observed in





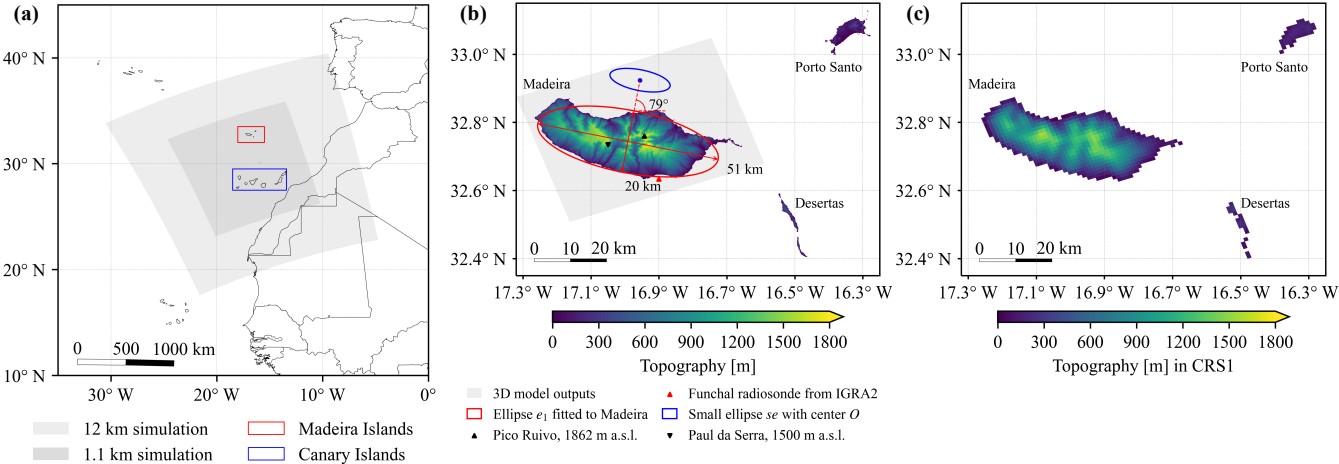

**Figure 1.** (a) 12 km (CPS12) and 1.1 km (CRS1) COSMO model simulation domains, and the location of Madeira and Canary Archipelagos. (b) Topography of Madeira Archipelago with 30 m horizontal resolution from European Environmental Agency under the GMES RDA project; The gray shading shows the region with decadal model outputs in all 60 (61) vertical model layers (levels). The Funchal radiosonde data from IGRA2 were used to calculate the inversion height. A red ellipse $e_1$ was fitted to the topography of Madeira using a Gaussian mixture model. The blue ellipse $se$ was obtained by scaling $e_1$ to its 1/9 in area and then shifting along its minor axis for the length of its minor axis. $se$ was employed to extract decadal 3D upstream conditions, which are only available within the gray shaded area in (b). It is important to mention that these upstream conditions cannot be assumed to be undisturbed due to the proximity of $se$ to the island. The altitude of two mountain peaks indicated by black triangles is used to compare with the inversion height. (c) The topography with a horizontal resolution of 1.1 km used in CRS1. The geographic map is from the Natural Earth.

the north or northeast windward regions than in the lee sides due to prevailing north-easterlies and orographic lifting (Cropper, 2013). The choice of Madeira Island for enhancing our understanding of land-atmospheric interactions is justified by its good isolation from continents, clear dynamical effects created by its topography, and the vulnerability to extreme weather events induced by its geographical conditions (Ramos et al., 2018).


Plenty of theoretical studies have been devoted to the dynamics of vortex streets and many observational studies concentrated on the geometry and kinematics. Nevertheless, they usually relied on limited case studies and idealized numerical simulations. This study makes a significant contribution to this research field with realistic simulations of the real atmosphere and a vortex identification algorithm. This algorithm allowed us to investigate the climatology of vortex shedding to the lee of Madeira.



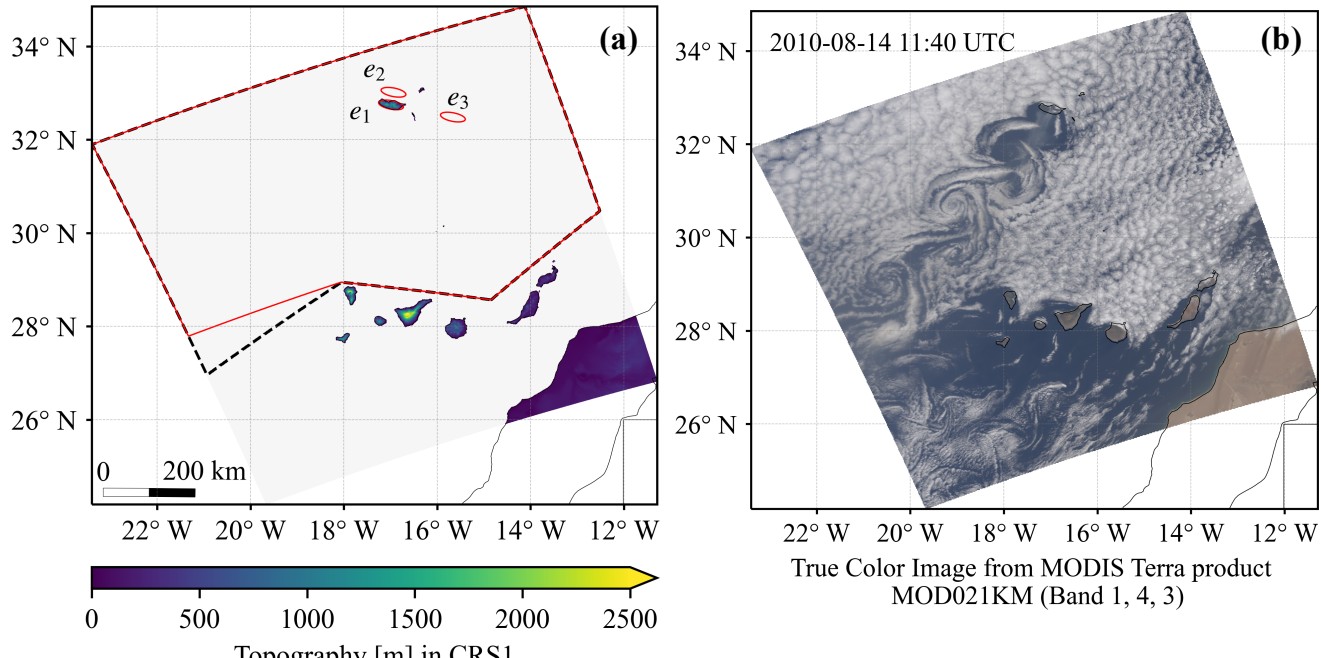

**Figure 2.** (a) The gray shading shows the region used for subsequent analysis. The red ellipses $e_2$ and $e_3$ were obtained by shifting $e_1$ as defined in Fig. 1b along its minor and major axes to extract upstream and undisturbed flow conditions, respectively. The black dashed and red polygons were used for vortex identification in Section 2.4. (b) An example of vortex streets to the lee of Madeira is visible through stratocumulus clouds using the data product MOD021KM from MODIS Terra satellite (Savtchenko et al., 2004). Such pictures can be viewed directly on the website of NASA Worldview (https://worldview.earthdata.nasa.gov). The geographic map is from the Natural Earth.

## 2 Data and methods

### 2.1 Model simulations

We used the non-hydrostatic Consortium for Small-Scale Modeling model in climate mode COSMO-CLM (Baldauf et al., 2011; Rockel et al., 2008). Specifically, we employed a refactored version of COSMO-CLM 5.09, which has been ported to run on hybrid GPU-CPU architectures in a joint effort between MeteoSwiss, the ETH-based Center for Climate Systems Modeling (C2SM), and the Swiss National Supercomputing Center (CSCS) (Fuhrer et al., 2014, 2018). In COSMO, a third-order Runge–Kutta split-explicit scheme is used for integrating compressible Euler equations (Wicker and Skamarock, 2002). A fifth-order upwind scheme is used for horizontal advection and an implicit Crank-Nicholson scheme is adopted for vertical advection. The model is equipped with a full set of parameterization schemes (see Ban et al., 2014). We used a horizontal resolution of 1.1 km for a convection-resolving simulation (CRS1) over east Macaronesia (1100 km × 1100 km) as shown in Fig. 1a. This simulation was run with the deep and shallow convection scheme off. Lateral boundary conditions were provided





by a convection-parameterizing simulation with a horizontal resolution of 12 km (CPS12) driven by the ERA-Interim reanalysis with 6-hourly frequency (Dee et al., 2011).

The simulation domain was defined relative to a rotated north pole at 43° N and 170° W, and contains 60 vertical layers up to 23.5 km altitude. The thickness of each layer widens from 20 m at the surface to around 1200 m near the model top. At the top

of the model domain, we used implicit Rayleigh damping for the vertical velocity following Klemp et al. (2008) to prevent the reflection of upward-propagating gravity-wave energy. CPS12 spanned from 2000-01 to 2015-12 including a spin-up period of 5 years, while CRS1 run from 2005-11 to 2015-12 and was initialized for 2 months before the analysis period to ensure a proper adjustment of the soil moisture (Ban et al., 2015). Both simulations provide hourly outputs on selected vertical levels over the entire respective simulation domains, and CRS1 also provides hourly data on all model levels over a smaller region

indicated by gray shadings in Fig. 1b. While the wind data are available at altitudes of 10 m and 100 m for the whole domain in CRS1, 100-meter winds at 1.1 km horizontal resolution were used for our vortex shedding analysis. For the case study in Section 3.2, we restarted the model simulation from 2010-08-01 00:00 UTC to generate 3D hourly data at a vertical interval of 100 m up to 5 km above the surface over the entire simulation domain of CRS1. Unless explicitly specified, the data used for analysis come from the original decadal CRS1.

Figure 1b shows the digital elevation model of Madeira Archipelago with a horizontal resolution of around 30 m based on data from the European Environment Agency (https://sdi.eea.europa.eu). We fit an ellipse centered at 32.74° N and 16.99° W with an area of about 800 $km^2$ to Madeira Island using a Gaussian mixture model (Reynolds, 2015). The length of the major and minor axes of the ellipse are about 51 and 20 km, respectively. The minor axis is oriented at 79° and is approximately parallel to the prevailing trade winds. Figure 2a shows the model domain of CRS1 used for analysis, where we excluded 80

grid points at the lateral boundaries to eliminate any effect of the relaxation zone. In addition to the original ellipse $e_1$ fitted to Madeira Island, we shifted it along its major and minor axes to extract upstream conditions at $e_2$ centered at 33.01° N and 16.94° W, and undisturbed flow conditions at $e_3$ centered at 32.48° N and 15.66° W. An example of vortex streets in the leeward of Madeira Island is shown in Fig. 2b.

## 2.2 Observational data sources

In addition to model simulations, we explored multiple observational data sources for validation or comparison. The ERA-Interim reanalysis (Dee et al., 2011), with a horizontal resolution of about 79 km and a frequency of 6 hours, was used to provide initial and boundary conditions for CPS12. For data analysis, we employed the ERA5 reanalysis (Hersbach et al., 2020), which is also produced by the European Centre for Medium-Range Weather Forecasts (ECMWF) and embodies hourly climate records at a higher horizontal resolution of around 31 km.

ASCAT is a C-band (5.255 GHz) fan-beam scatterometer system on polar orbiting meteorological operational (MetOp) platforms from the European Organisation for the Exploitation of Meteorological Satellites (EUMETSAT) (Figa-Saldaña et al., 2002). The primary mission of ASCAT is to provide 10-meter global ocean wind vectors that are mainly used for assimilation in numerical weather prediction models. On ASCAT, six antennas cover two 550 km swaths with an interval of 360 km at a minimum orbit height of 822 km. A wind product, at a resolution of 25 km on a 12.5 km nodal grid, is generated from ASCAT



level 1b data using an empirical geophysical model function by the Ocean and Sea Ice Satellite Application Facility (OSI SAF). However, we used ASCAT Wind Data Processor (AWDP) software to derive wind data from measurements of ASCAT onboard the Metop-A satellite at a higher resolution of around 17 km on a 6.25 km grid to provide more details about vortex streets (Vogelzang et al., 2017). Note that although the terms horizontal resolution and grid spacing are often used interchangeably (also in our study), they are different for ASCAT wind products and the products introduced below, as their values at each grid

are derived by aggregating observations from surrounding grids.

     For the comparison of vortex streets in simulations and the real atmosphere, we employed the high rate SEVIRI level 1.5 image data of the geostationary satellite Meteosat-9 in the Meteosat Second Generation (MSG) from EUMETSAT (Schmetz et al., 2002). These data provide high rate transmissions in 12 spectral channels covering 81° S to 81° N and 79° W to 79° E every 15 minutes. They have a sampling distance of 3 km and a horizontal resolution of 4.8 km at the sub-satellite point, and

a sampling distance of 4 to 5 km in our analysis region. An introduction to create and interpret standard RGB images from METEOSAT/SEVIRI can be found on the website of an international training project EUMeTrain founded by EUMETSAT (https://www.eumetrain.org). We used the Natural Colors RGB for daytime (08:00 - 19:00 UTC) and the Night Microphysics RGB for nighttime.

     Vertical temperature profiles from radiosonde observations at Funchal and Tenerife on a daily (normally at 12:00 UTC) and

twice-daily (normally at 00:00 and 12:00 UTC) basis, respectively from the Integrated Global Radiosonde Archive (IGRA) version 2 (Durre et al., 2006) were explored for a decadal climatology of inversion height. The location of Funchal radiosonde station is shown in Fig. 1b. As the geopotential height is often not available at reported pressure levels, it was computed using the hydrostatic relation (Durre and Yin, 2008). Albeit IGRA V2 provides the inversion height defined as the height of the warmest temperature in the sounding, we estimated it based on the definition from Kahl (1990), i.e. the lowest layer where

temperature increases with height.

     To validate simulated precipitation patterns, we used a data set of conventional meteorological surface stations. The network contains 17 stations in the height range 25-1799 m. Most stations cover at least 5 years of the simulation period 2006-2015. The validation of precipitation against this data set is conducted following the method described in Ban et al. (2020). That is, CPS12 model outputs are interpolated to station sites with nearest-neighbor interpolation and for CRS1 the grid cell with the

smallest altitudinal difference within a 4 km radius is selected for comparison.

## 2.3   Basic flow parameters

Following Snyder et al. (1985) and Heinze et al. (2012), the Froude number $Fr$ was defined here as the ratio of inertial to buoyancy forces. In the case of a uniformly stratified airstream, it is given by $Fr = U/(Nh_m)$ where $N$ denotes the Brunt-Väisälä frequency, $U$ the upstream wind speed, and $h_m$ the mountain height. In a number of studies, the inverse of $Fr$ was

alternatively used and referred to as the dimensionless mountain height, i.e. $h_{dim} = Fr^{-1} = Nh_m/U$.

     The concept of the dividing streamline $h_{ds}$ is commonly used in the analysis of flow past 3D topography. Here the height of this streamline $h_{ds}$ is the upstream height that separates the streamlines passing over a three-dimensional hill in a stably stratified flow from those passing around the hill. $h_{ds}$ can be calculated based on Sheppard's energy theory (Sheppard, 1956):





upstream air parcels at an altitude above $h_{ds}$ would have sufficient kinetic energy to surmount the island, whereas below $h_{ds}$
the flow is quasi horizontal around the island. Sheppard's energy theory assumes that all kinetic energy is converted to potential
energy, and thus $h_{ds}$ provides the lowest possible dividing streamline height. In the case of general upstream profiles $U(z)$ and
$N(z)$, $h_{ds}$ can be estimated from (Heinze et al., 2012)

$$\frac{1}{2}U^2(h_{ds}) = \int_{h_{ds}}^{h_m} N^2(z)(h_m - z)dz\,, \tag{1}$$

where $N(z)$ is the Brunt-Väisälä frequency defined as

$$N(z) = \left(\frac{g}{\theta(z)}\frac{\partial\theta(z)}{\partial z}\right)^{1/2}. \tag{2}$$

Here $g = 9.81 \text{ m s}^{-2}$ is the gravitational acceleration and $\theta(z)$ the upstream potential temperature.

For upstream profiles of uniform $N$ and $U$, $h_{ds}$ can be obtained analytically as (Sheppard, 1956)

$$h_{ds} = h_m(1 - Fr)\,. \tag{3}$$

In atmospheric flows past 3D obstacles, vortex shedding occurs at $Fr < 0.4$ (Etling, 1989), which appears to be roughly con-
sistent with the laboratory experiments of Hunt and Snyder (1980). These results are also consistent with idealized numerical
experiments using free-slip lower boundary conditions (Schär and Durran, 1997). For instance, Schär and Durran (1997) found
vortex shedding for $Fr = 1/3$, while for $Fr = 2/3$ there was nonlinear flow over the topography with gravity wave breaking
and no vortex shedding.

However, the validity of Sheppard's energy theory was questioned by Smith and Gronas (2016) from a theoretical perspec-
tive. They argued that the deceleration of air parcels approaching the mountain is due to the pressure gradient associated with
the hydrostatic gravity wave aloft, rather than due to vertical displacements. However, we believe that in situations with a
pronounced inversion, as considered here, the dividing streamline concept is valid.

In our realistic simulation, $N$ and $U$ vary vertically, so we approximated Eq. (2) as

$$N^2 = \frac{g}{\theta}\frac{\theta(h_m) - \theta(h_{surf})}{h_m}\,, \tag{4}$$

for the estimation of $h_{dim}$, where $h_{surf}$ denotes the surface height. Likewise, $U$ and $\theta$ are expressed as the vertical average of
wind velocity and potential temperature below $h_m$, respectively.

The Rossby number is a dimensionless parameter for rotating reference frames. It is defined by the ratio between inertial
force and Coriolis force as

$$Ro = \frac{U}{fL}\,, \tag{5}$$

where $L$ is the relevant spatial scale, and $f = 2\Omega\sin\phi$ is the Coriolis parameter with $\Omega = 7.29 \times 10^{-5} \text{ s}^{-1}$ being the rate of
Earth's rotation and $\phi$ being the latitude.





Another important parameter, defining the dimensionless shedding frequency of vortices, is the Strouhal number, defined as

$$St = \frac{D}{TU}, \tag{6}$$

where $D$ is the crosswind island diameter at the inversion base, and $T$ is the shedding period between two successive vortices
on the same side of obstacles.

## 2.4 Rule-based vortex identification

In this section, we introduced a rule-based vortex identification algorithm consisting of wavelet analysis and a series of criteria. The central idea of our algorithm inherited from McWilliams (1990). He suggested judging the reliability of an algorithm in terms of both the prior plausibility of criteria and the posterior performance.

Several concerns in designing criteria are illustrated below. To extract individual vortices, we firstly denoised the spatial field of relative vorticity using wavelet analysis (Doglioli et al., 2007), of which theoretical and technical aspects are presented in Appendix A. Although in the case of ocean eddies, it is more objective to determine vortex boundaries based on closed streamlines (Nencioli et al., 2010), this method does not apply to atmospheric vortices due to strong background winds. As older shedded vortices have a smaller area with relative vorticity above a given threshold, the main challenge is to separate
small vortices from noise. Accordingly, we intentionally put stricter constraints on smaller vortices. The selected criteria are presented and explained below, and the order is designed to maximize computational efficiency.

1. Minimum magnitude of absolute values of relative vorticity for a vortex at a height of 100 m: $3 \times 10^{-4}\,\mathrm{s}^{-1}$.

   This step identifies grid cells that could belong to a vortex and clusters adjacent grid points above a threshold. The cell identification is based on a cloud identification algorithm from Mosimann (2016). Mosimann (2016) used a list-based
union-find data structure (e.g., Cormen et al., 2001) to successively build up the clusters.

2. Minimum peak magnitude of absolute values of relative vorticity in a vortex: $4 \times 10^{-4}\,\mathrm{s}^{-1}$.

   This criterion poses a weak constraint on peak absolute values of relative vorticity in extracted clusters from the first step.

3. Minimum size of a vortex: $100\,\mathrm{km}^2$.

Albeit this criterion will eliminate smaller vortices, it is an essential step to distinguish vortices from noise.

4. A vortex should not be in contact with the model topography.

   If a vortex contacts with the model topography, we regard it as not shedded, although it might be in its generation phase.

5. The center of a vortex should be located within the black dashed polygon in Fig. 2a for our case study in Section 3.2 and within the red polygon in Fig. 2a for our climatological study in Section 3.3.

The black dashed polygon denotes the region where vortices shedded from Madeira Island can appear and was used for the case study to identify as many vortices as possible. However, if it was employed for decadal vortex identification, a



significant number of vortices shedded from La Palma island in the Canary Archipelago would be wrongly incorporated. So for decadal analysis, we adopted the red polygon to avoid the interference of La Palma island.

6. If the vortex size is less than $450 \text{ km}^2$, there must be a local potential temperature maximum at 2 m altitude within the vortex area or within the circle defined by the nominal radius around its center.

This criterion is well justified by the properties of shedded vortices. As reported by Heinze et al. (2012), both positive and negative shed vortices have a warm core and do not exhibit a significant vertical tilt within the boundary layer. This structure is also supported in Fig. 12 and Fig. 13. This feature was used to track vortex shedding in idealized simulations (Heinze et al., 2012). The nominal radius of a vortex is defined as the radius of a circle that has the same area as the vortex.

7. If the vortex size is in the range of less than 450 (450 to 900, or larger than 900) $\text{km}^2$, the angle between the mean wind vector within a vortex and the vector connecting the Madeira center and vortex center should be less than 30° (40°, or 50°).

This requirement restricts the advection direction of a vortex. The thresholds were selected conservatively based on our observations to exclude vortices that are unlikely to be shedded from Madeira Island. Although shedded vortices can be deviated from their original tracks by synoptic flows (Etling, 2019), we consider this condition as an exception and focused on normal cases.

8. Maximum ratio of the largest distance between two grid cells within a vortex to its nominal radius: 5.

The criterion limits the deviation of vortex shape from a circle (McWilliams, 1990). It is mainly used for the elimination of elongated vorticity signals, which may be due to noise or vorticity banners attached to the terrain (Schär et al., 2003).

9. There must be at least one vortex satisfying all above conditions either at the previous hour or the next hour.

It aims at excluding isolated vortices that have no precursor and successor.

## 3 Results

### 3.1 General meteorological conditions

Fig. 3a presents annual mean precipitation over our analysis region in the ERA5 reanalysis from 2006 to 2015. It ranges from 43 to 564 mm with increasing amounts from the southeast to the northwest. Wetter conditions can be observed near the islands as an effect of orographic lifting. Although Fig. 3a-c indicate negative biases of simulated precipitation in comprison to the ERA5 reanalysis over the ocean, especially in CRS1, the above-mentioned spatial patterns are well simulated in both CPS12 and CRS1. Fig. 3d-f display precipitation over Madeira Archipelago in a zoomed-in version. The horizontal resolutions of the ERA5 reanalysis (0.25°) and CPS12 (12 km) are insufficient to resolve the topography. Indeed, maximum elevation over Madeira amounts to around 1862 m in reality, 1546 m in CRS1, 600 m in CPS12, and 330 m in ERA5. As a result, the lifting





and splitting of the ambient flow cannot properly be represented in ERA5 and CPS12, yielding too small precipitation amounts and unrealistic patterns. In contrast, CRS1 produces much larger precipitation amounts approximately centered over Madeira.

The high values over Madeira Island in CRS1 are qualitatively consistent with the fact that occasionally even flash floods can
occur (Fragoso et al., 2012; Luna et al., 2011). To quantitatively validate the simulations, we compared simulated annual mean precipitation against observations at conventional surface stations (Fig. 4). The CPS12 simulation reaveals a large underestimation with typical biases amounting to 40-80%, in particular at mountain stations. In CRS1 we found more realistic results as the convection is partially resolved and the topography is more realistic. In particular, some of the mountain stations now exhibit precipitation amounts around 1000 mm close to observations, and the biases are reduced considerably. However, precipitation
is still underestimated, and the asymmetry of the Madeira precipitation (with higher amounts upstream) is not fully captured. It is likely that these systematic errors are due to the representation of microphysical processes, as a considerable fraction of precipitation in Madeira falls as drizzle.

Figure 5 demonstrates annual variations in precipitation over Madeira Island in the ERA5 reanalysis, CPS12, and CRS1. Although the grid cells used in each dataset do not cover the same region (see the caption), the comparison reveals that three
datasets qualitatively agree. Interestingly, a drying trend can be observed in both observations and simulations from April to August, and is interrupted by a sudden change to wetter conditions in September. Although there is no study reporting a direct relationship between precipitation and vortex shedding, we can see a synchronous change in vortex shedding patterns in Section 3.3.

The 100-meter wind climatology in CRS1 during 2006-2015 over the analysis region defined in Fig. 2a is shown in Fig. 6.
Equivalent plots with data from the ERA5 reanalysis (not shown) have quite similar monthly patterns and indicate good performance of our simulation in terms of wind climatology. Our analysis region is dominated by north-easterlies throughout the year, consistent with trade winds. A significant concentration in wind direction and an increase in mean wind speed from April to August generate favorable conditions for vortex shedding as shown in Section 3.3.

In Fig. 7a, the monthly inversion height observed at the Funchal radiosonde station from 2006 to 2015 also shows very
interesting patterns. Firstly, the observed monthly mean inversion height decreases from about 1.5 km in April to around 1 km in July and August, which is conducive for flow splitting and thus vortex shedding. It then increases to above 1.6 km in September and results in unfavorable conditions for vortex shedding, like the winds in Fig. 6, both of which are influenced by the Azores high-pressure system. The observed monthly mean inversion height is about 1.5 km from September to April. By comparing the monthly inversion height in observations and simulations in CRS1, we can detect that their standard deviations
are rather close and the monthly patterns are very well simulated despite consistent negative biases of around 150 m. As atmospheric conditions at Funchal are disturbed by Madeira Island, we also showed the monthly inversion height in CRS1 over the ellipse $se$ from Fig. 1b with upstream conditions under prevailing north-easterlies. Unfortunately, the flow in $se$ is still strongly affected by the orography and therefore does not represent undisturbed upstream conditions (which we lack due to the limited domain with 3D data). However, using it still gives us some indications about the variability of the inversion height.
The simulated values over $se$ are close to those near Funchal except in July and August, where the monthly mean inversion





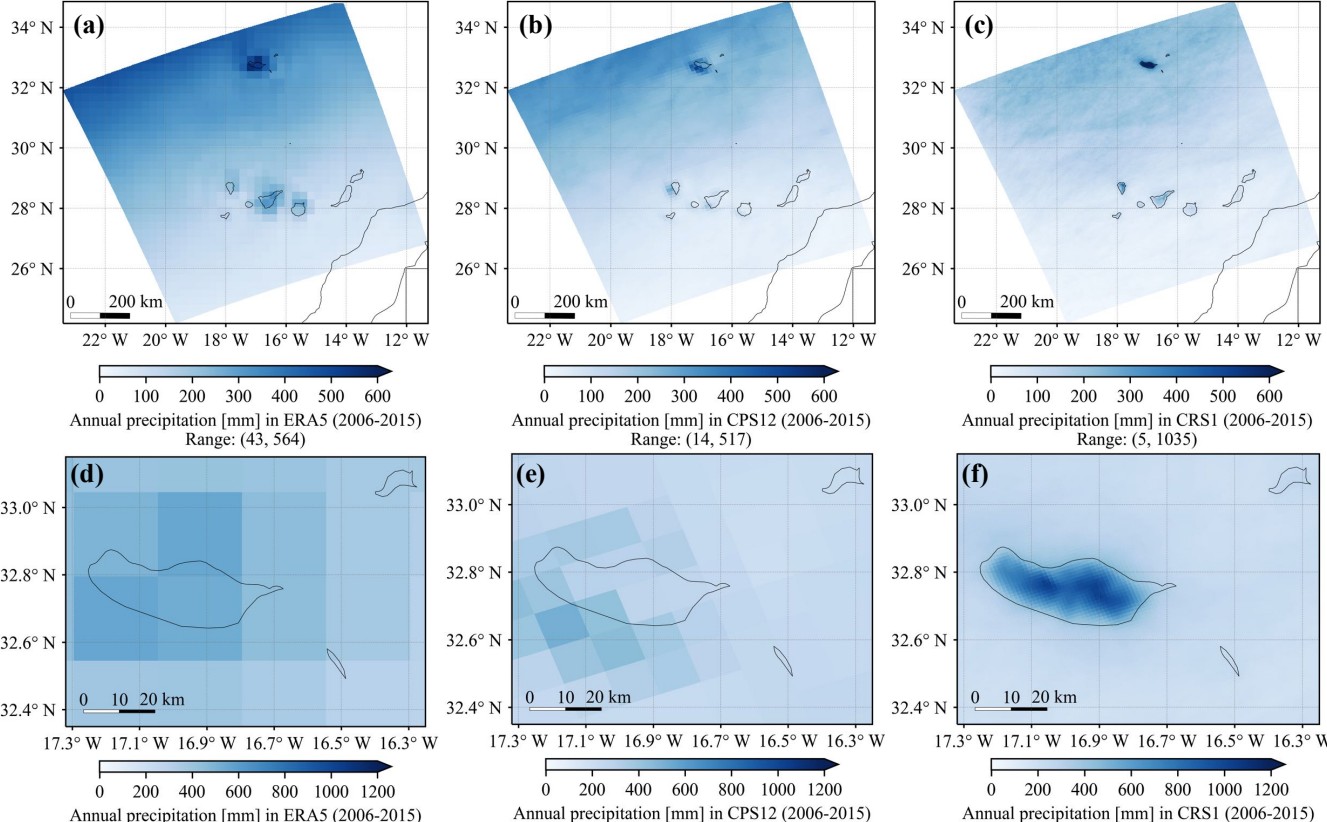

**Figure 3.** Annual precipitation [mm] over (a-c) the analysis region and (d-f) Madeira Archipelago in (a, d) the ERA5 reanalysis, (b, e) CPS12, and (c, f) CRS1 from 2006 to 2015. Note the different scales for the top and bottom row panels. The geographic map is from the Natural Earth.

height is lower at Funchal, possibly due to downslope winds (Durran, 1990) resulting from strong north-easterlies as shown in Fig. 6.

The simulation data for Fig. 7a were extracted using the same timestamps as the Funchal radiosonde observations with 98.4% of the observations being conducted at 12:00 UTC. Consequently, we present the diurnal cycle of inversion height in Fig. 7b to examine the representativeness of the data at 12:00 UTC. The model outputs near Funchal show a clear diurnal cycle. The hourly mean inversion height exceeds 1250 m from 12:00 to 17:00 UTC and lies below 1150 m from 21:00 to 08:00 UTC. Such a diurnal cycle can also be observed for each calendar month (not shown). Hence, the observations and simulations near Funchal primarily at 12:00 UTC in Fig. 7a are representative even though the 12:00 UTC data exhibit around 80 m higher inversion height than the daily mean. As the simulated values over ellipse $se$ in Fig. 7b display nearly no diurnal cycle, the simulated values over ellipse $se$ in Fig. 7a are also quite representative. The inversion height estimated with the Tenerife



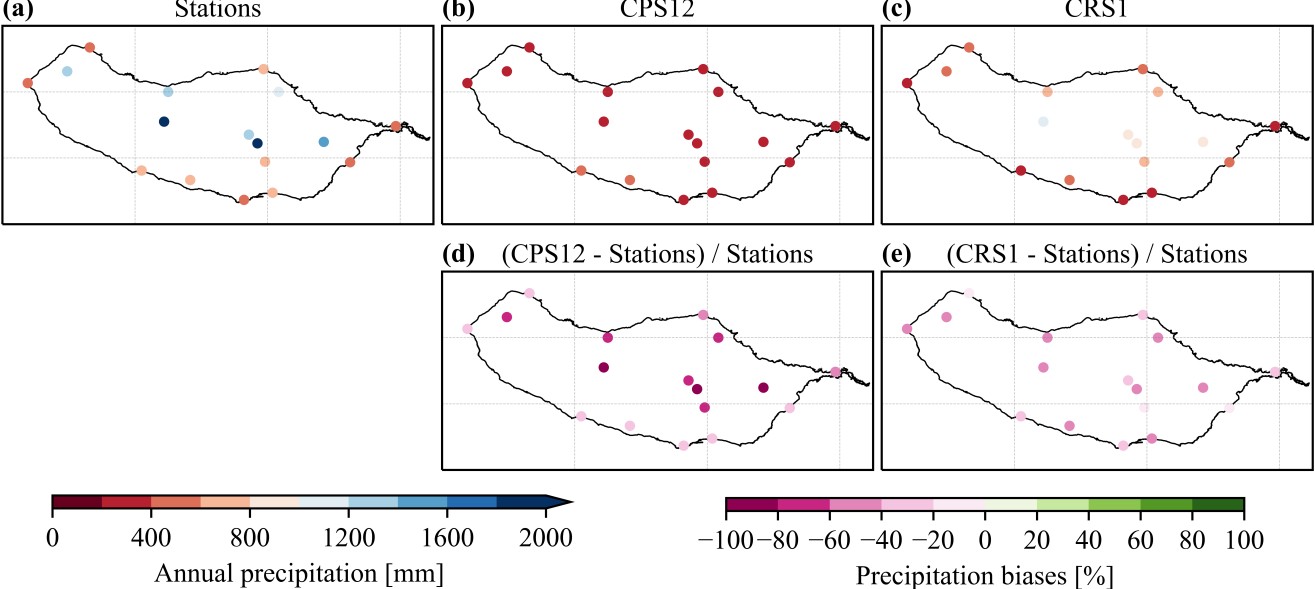

**Figure 4.** Validation of precipitation [mm] against conventional surface stations. Annual-mean precipitation amounts (2006-2015) from (a) the observational dataset, (b) CPS12, and (c) CRS1. Relative biases [%] of (d) CPS12 and (e) CRS1 with respect to observations at stations. The geographic map is from the GADM.

radiosonde observations at 28.32° N and 16.38° W reveals similar monthly patterns and negative biases in simulations as in Fig. 7a, and is generally lower than that at Funchal (not shown).

### 3.2 Case study of 03-09 August 2010

To investigate the characteristics of vortex streets, we selected a vortex shedding event that lasted for about one week from
2010-08-03 20:00 UTC to 2010-08-09 07:00 UTC. The animation of simulated 100-meter wind velocity and direction during this period can be found in Supporting Information V1 (Gao et al., 2021). The average upstream wind velocity is around 12 m s$^{-1}$. We can see that winds are accelerated at island flanks and decelerated before and behind Madeira Island, which indicates horizontal flow splitting. The corresponding animation of derived 100-meter relative vorticity is shown in Supporting Information V2 (Gao et al., 2021), which acts as a good tracer for vortex streets and shows some variations during this
event. In animation V2, lime contours encompass 347 identified vortices using our vortex identification algorithm described in Section 2.4. For this case study, the vortex detection methodology was checked manually. Of all identified vortices, 344 were identified correctly, and three were wrongly identified and highlighted in cyan in animation V2. This corresponds to a false-positivity rate (FPR) (Yang et al., 2020) of 0.9% and one falsely identified vortex every 44 hours. Occasionally, stratocumulus clouds can also serve as tracers for vortex streets, so we visualized cloud patterns during this period in Supporting Information
V3 (Gao et al., 2021) using the high rate SEVIRI rectified image data of the Meteosat-9 in MSG from EUMETSAT (Schmetz




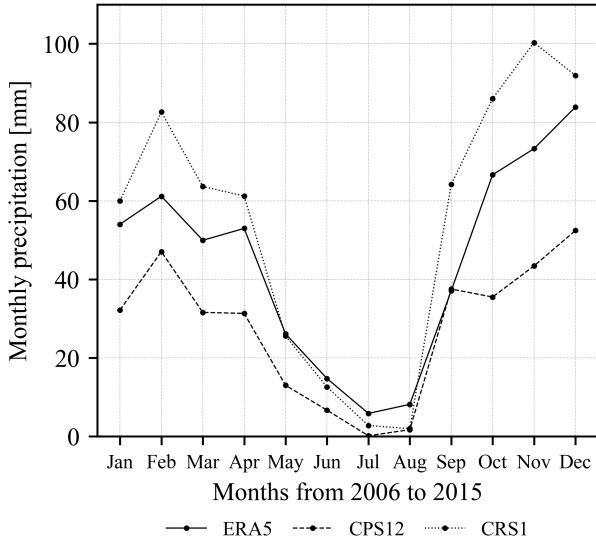

**Figure 5.** Monthly precipitation [mm] over Madeira Island in the ERA5 reanalysis, CPS12, and CRS1 from 2006 to 2015. For each dataset, the grid cells whose center are located within the ellipse $e_1$ in Fig. 2a were used to calculate the spatially-averaged monthly precipitation. That is, one grid cell ($\sim$961 km$^2$) in the ERA5 reanalysis, six grid cells ($\sim$864 km$^2$) in the CPS12, and 666 grid cells ($\sim$800 km$^2$) in the CRS1 were adopted.

et al., 2002). Although cloud patterns are not as indicative as relative vorticity for the detection of vortex streets, we can still observe a well-organized vortex shedding event in animation V3 that agrees well with the simulations.

Figure 8 demonstrates time series of local flow conditions around Madeira Island during this period. The uppermost array of arrows represents mean 100-meter wind direction and velocity within ellipse $e_3$ in Fig. 2a, which can be assumed as undisturbed
flow conditions. Wind velocity in $e_3$ generally exceeds 10 m s$^{-1}$ with only slightly varying wind direction. Winds over $e_2$ are much weaker and significantly dispersed, so we used undisturbed flow conditions from $e_3$ for subsequent analysis. The green line in Fig. 8 shows the number of correctly identified vortices shedded from Madeira Island, which shows large variations during this period. The dimensionless mountain height $h_{dim}$ within $e_3$ shown in the red line is generally above 3, a value that Schär and Durran (1997) used to specify flow conditions favorable for vortex shedding. The dividing streamline height $h_{ds}$
mostly exceeds 60% of the peak mountain height in CRS1 ($\sim$0.93 km denoted with a dotted black line), which is a widely used threshold to distinguish flow conditions conducive for vortex shedding. The weak vortex shedding signals at the middle of this period seems coincide with stages where $h_{ds}$ and $h_{dim}$ fall below the corresponding thresholds. In addition, a sudden increase of the inversion base over $e_3$ in the solid blue line at 2010-08-06 05:00 UTC also precedes the increasing vortex shedding signal. The comparison of inversion height from Funchal radiosonde observations at 12:00 UTC and a grid cell in CRS1 closest to
Funchal radiosonde station in the dashed and dotted blue lines shows expected patterns as described in Section 3.1: negative bias and vulnerability to perturbations in lower levels in the simulations.



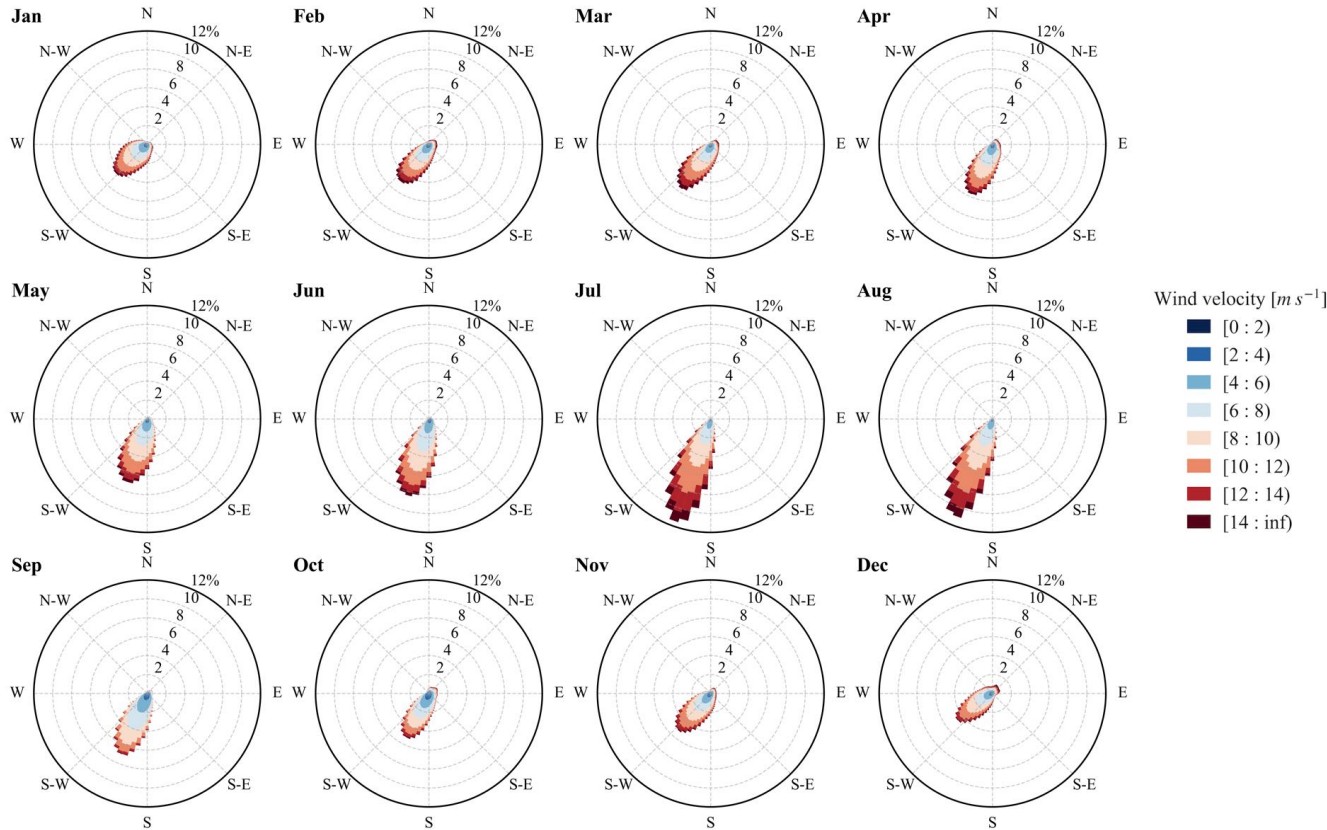

**Figure 6.** Simulated monthly 100-meter wind climatologies for 2006-2015 in CRS1 over the analysis region defined in Fig. 2a based on hourly data in CRS1. The wind roses can be understood as histograms of wind direction that simultaneously take into account wind velocity varying from 0 to more than 14 $\text{m s}^{-1}$. Each sector with a width of $5°$ points in the direction of the wind and its radius represents the frequency.

Synoptic flow conditions including mean sea-level pressure and 10-meter winds from the ERA5 reanalysis during this period are shown in Fig. 9. At the beginning of this period, a high-pressure system centered north of the Azores resulted in a strong north-easterly flow, especially over the northwest of our study domain. Subsequently, the high-pressure system moved northeast and deformed towards a circle, which led to a stronger north-easterly flow that stimulated well-organized vortex streets in the lee of Madeira Island, whose major axis is approximately perpendicular to the mesoscale wind direction. At the end of this event, the vortex shedding phenomenon weakened with the vanishment of the high-pressure system. Consequently, we can identify a clear correlation between vortex shedding in the lee of Madeira Island and the high-pressure system, which is well known as the semipermanent Azores high (Davis et al., 1997) and forms one pole of the North Atlantic Oscillation (NAO) (Visbeck et al., 2001).



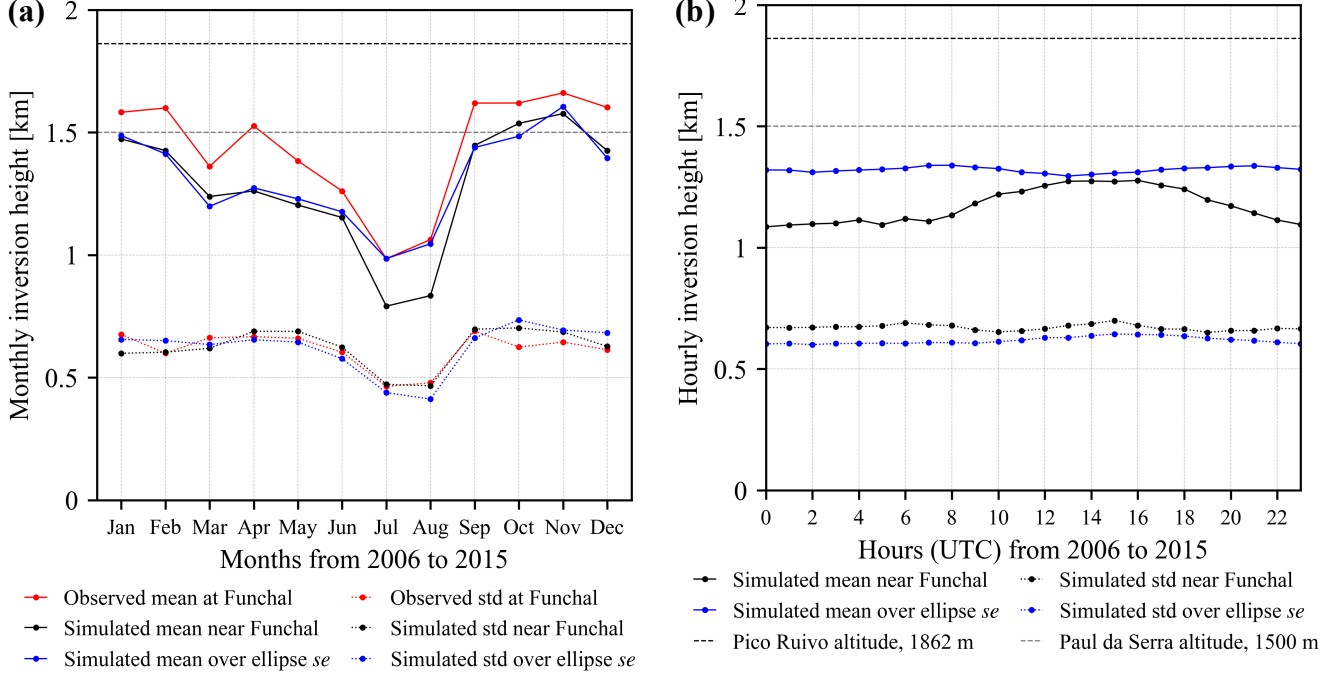

**Figure 7.** (a) Monthly climatology of inversion height and (b) diurnal cycle of inversion height. Both panels show monthly/hourly means and standard deviations (std). The observed values were calculated from the Funchal radiosonde data from IGRA2, and the simulated values near Funchal were calculated based on the model simulation CRS1 at the grid point closest to the Funchal radiosonde station. The simulated values over $se$ were used to represent upstream conditions where decadal 3D model outputs are available. The altitude of two mountain peaks, Pico Ruivo (1862 m) and Paul da Serra (1500 m) at Madeira Island, are indicated for reference.

We manually tracked four positive vortices labeled as P1 to P4 and four negative vortices (N1-N4) starting from the selected timestamp. Tracks of P1 and N1 are shown in Fig. 10 and the animation of their propagation can be found in Supporting Information V4 (Gao et al., 2021). Based on the eight vortices, we calculated several key numbers to compare with existing literature and satellite images. The ratio of advection velocity of vortices to undisturbed wind velocity over $e_3$ has a mean of

0.82 and a standard deviation of 0.02, while existing literature reported a large range of values like 0.76 in Tsuchiya (1969) or $0.90 \pm 0.37$ in Heinze et al. (2012). The average shedding interval of like-rotating vortices is about 6.7 hours, and a rough estimation based on satellite images in this period gives a value of around 7 hours. The corresponding Strouhal number has a mean of 0.154 and varies from 0.142 to 0.166, which also lies in or close to the reported range from Nunalee and Basu (2014) between 0.15 and 0.22. Figure 11 presents the size of eight tracked vortices from $100 \, \mathrm{km}^2$ to $1600 \, \mathrm{km}^2$ against their distance

to Madeira center up to $600 \, \mathrm{km}$. We can see a faster decay of negative vortices, which might be attributed to the Coriolis force, as the Rossby number defined in Eq. (5) has a value of around 4 considering a relevant length scale of $L = 35 \, \mathrm{km}$, a flow velocity of $u = 10 \, \mathrm{m \, s}^{-1}$, and a latitude of $\phi = 32 \, \mathrm{deg}$.





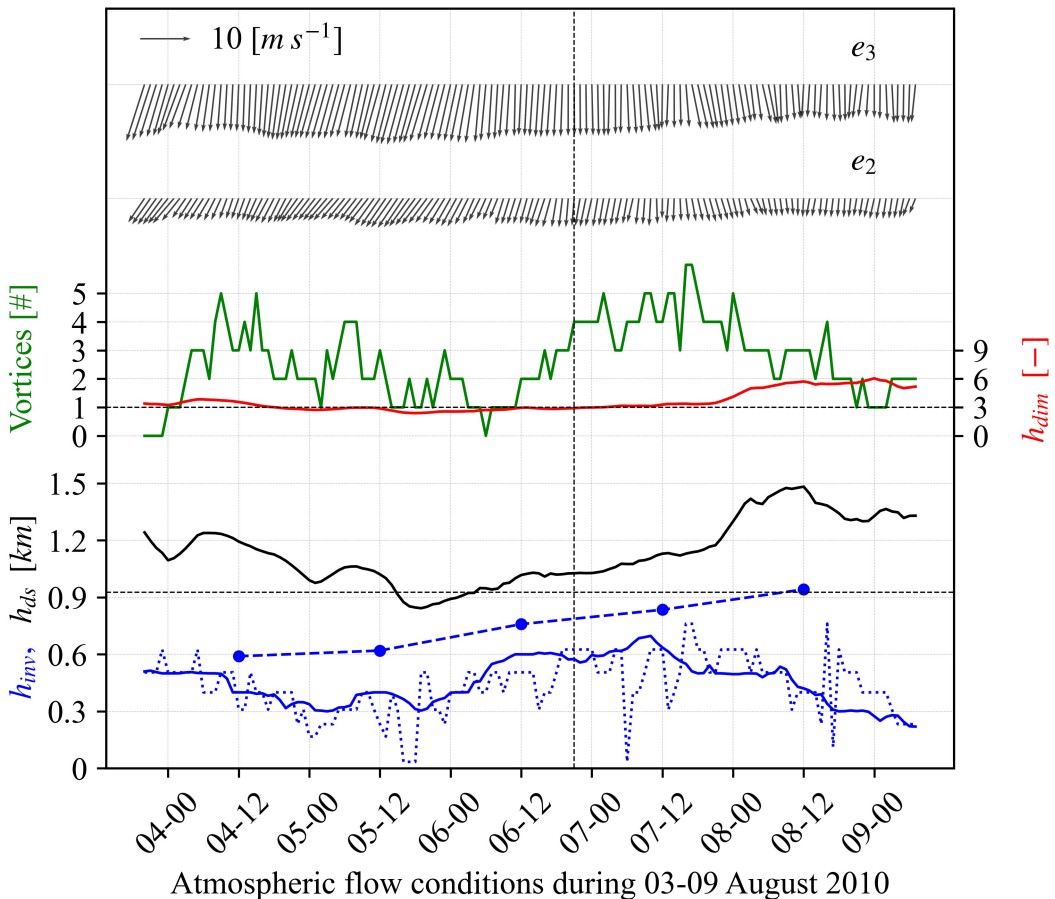

**Figure 8.** Local flow conditions during the case study period (from 2010-08-03 20:00 UTC to 2010-08-09 07:00 UTC). The top two rows of arrows represent average wind arrows over ellipses $e_3$ and $e_2$, and a wind velocity of $10 \, \text{m s}^{-1}$ is shown for reference. The third row shows the number of identified vortices shedded from Madeira (green line), and the average dimensionless mountain height $h_{dim}$ over $e_3$ (red line). The dividing streamline height $h_{ds}$ over $e_3$ are shown in black line. The solid blue line represents the inversion height $h_{inv}$ over $e_3$, and dashed and dotted blue lines show $h_{inv}$ in Funchal radiosonde observations and CRS1 in a grid cell closest to the Funchal radiosonde station, respectively. A selected timestamp 2010-08-06 21:00 UTC for subsequent analysis is indicated by a vertical dashed black line.

For the decade-long simulations, only a reduced set of output fields was stored. To investigate the simulated vortex streets in detail, we restarted the model simulation on 2010-08-01 00:00 UTC for the case study period to generate variables every 100 365 m in the lowest 5 km above the surface. Figure 12 shows the 100-meter relative vorticity, 10-meter potential temperature, and 100-meter specific humidity at the selected timestamp in the new model simulation. Small differences can be found between Fig. 12a and Fig. 10a due to stochastic properties of simulations. We can identify positive potential temperature anomalies at 10 m altitude as mentioned in Section 2.4 and positive specific humidity anomalies around both positive and negative vortices. Both kinds of anomalies were observed in the wake of Hawaii (Smith and Grubišić, 1993). The potential temperature anomalies



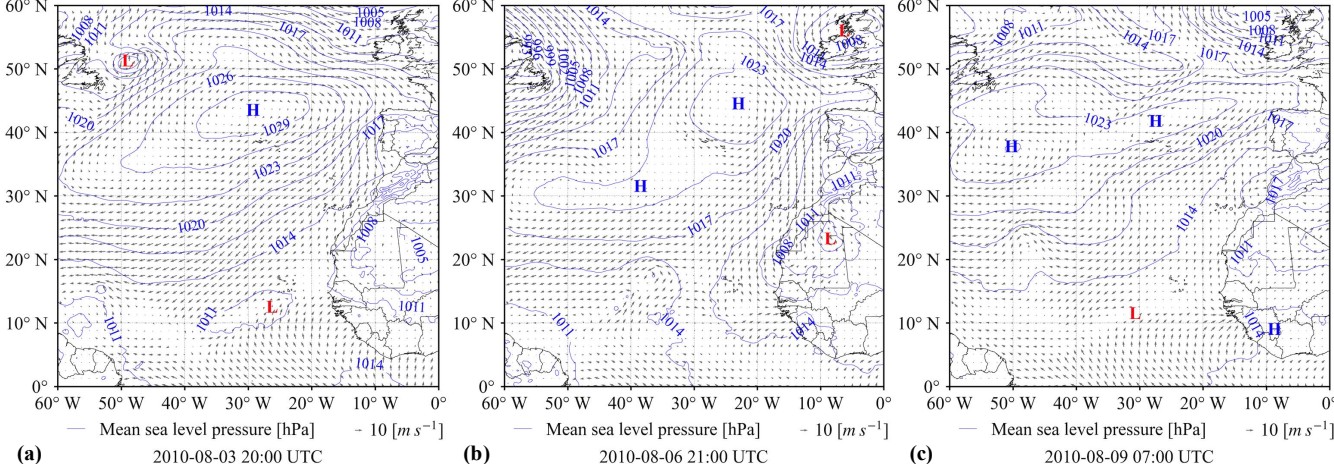

**Figure 9.** Synoptic flow conditions including mean sea level pressure and 10-meter winds from the ERA5 reanalysis at the (a) starting, (b) middle, and (c) ending time of the case study period. A wind velocity of $10 \mathrm{~m \, s^{-1}}$ is shown for reference. H (L) represents high (low) pressure systems. The geographic map is from the Natural Earth.

370 result mainly from temperature anomalies and could be used as a tracer for vortex streets in idealized simulations (Heinze et al., 2012).

  The vertical structures of vortex streets along two cross-sections $C_1$-$C_2$ and $C_3$-$C_4$ depicted in Fig. 12a are shown in Fig. 13. The two cross-sections show similar signals. From Fig. 13a we can observe a well-mixed upstream boundary layer, gravity waves above the island, distorted downstream isentropes, a lowering of the downstream capping inversion, and six downstream

375 positive potential temperature anomalies extending from the surface to the inversion height. The last potential temperature anomaly from $C_1$ to $C_2$ has no corresponding vortex, which indicates that vortex streets can still impact the atmosphere further downstream even though the vortices already dissipated. The lifting condensation level (LCL) is the height where an air parcel would saturate when lifted adiabatically. Although the LCL is estimated based on surface data (Romps, 2017), it well captures the lower bounds of contours of 95% relative humidity. The cloud-free region in the lee of Madeira Island in the animation V3

380 in Supporting Information (Gao et al., 2021) can be explained by the lowering of the inversion base below the LCL and can result in intense solar radiation and thus a warm oceanic wake (Azevedo et al., 2020). In satellite images, we could sometimes observe cloud-free conditions (eyes) in the center of shedded vortices. The lower inversion base than LCL was assumed to be responsible for cloud-free eyes of vortices (Heinze et al., 2012), but this is not supported by our simulation as the inversion base is higher than LCL at downstream vortices.

385 It is still difficult to validate the track and timing of individual vortices in realistic simulations because of limited observations. During our case study period, vortex streets in the lee of Madeira Island were captured in only three snapshots by ASCAT onboard the MetOp-A satellite, of which the clearest one at around 2010-08-05 10:40 UTC are shown in Fig. 14a-b. The 10-meter relative vorticity derived from the observed wind vectors shows distinct vortex shedding signals which coincide with



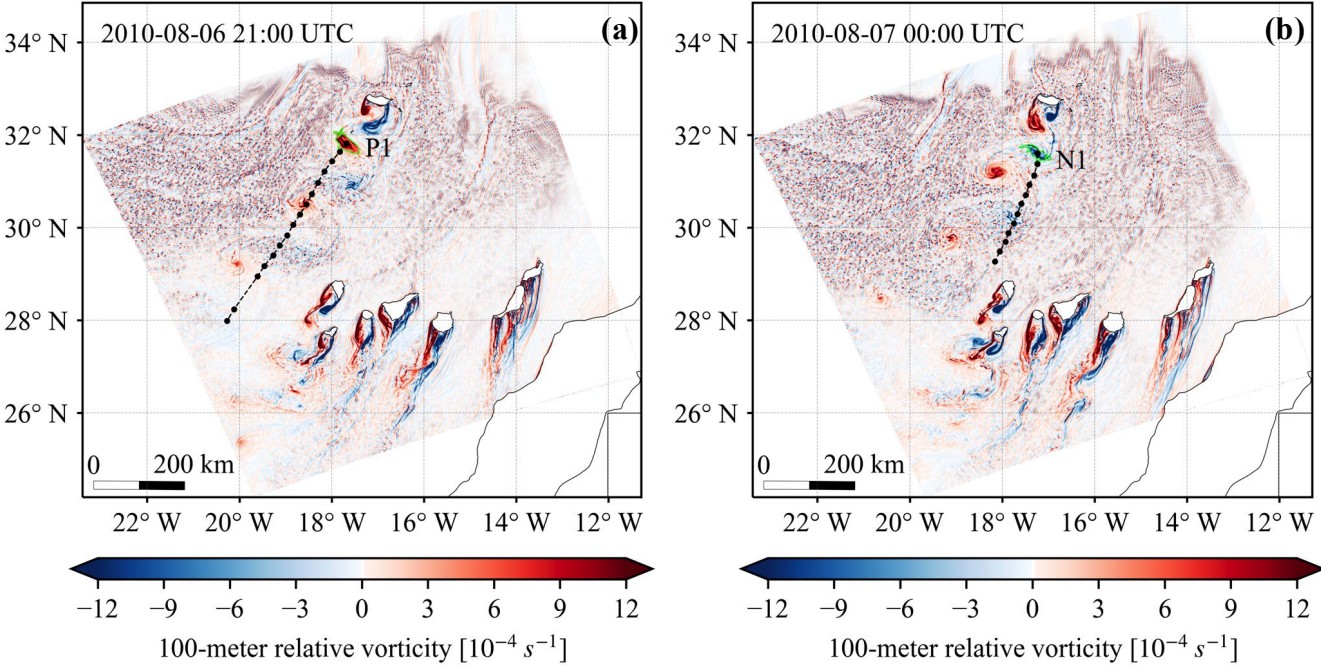

**Figure 10.** Relative vorticity at 100 m and tracks of (a) a positive vortex P1 shedded at 2010-08-06 21:00 UTC and (b) a negative vortex N1 shedded at 2010-08-07 00:00 UTC. The relevant vortices are encompassed by lime contours, and the tracks in next hours are shown with black lines and dots on an hourly basis. A gap in the lower half of the track of P1 results from a false rejection of that vortex by our algorithm. The geographic map is from the Natural Earth.

cloud patterns observed by Meteosat-9 in Fig. 14c. The corresponding model outputs at 2010-08-05 11:00 UTC in Fig. 14d-e

also show obvious vortex streets in the lee of Madeira Island, which agrees well with the observations. Note also that we cannot expect a perfect spatial fit as vortex shedding is a chaotic phenomenon and the timing in a long simulation such as ours. The simulated cloud patterns in Fig. 14f do not agree with the observations and it looks like the model has difficulties in the simulation of clouds for this specific situation. This is not uncommon, as the simulation of clouds is still a big challenge even in high-resolution weather and climate models (Bretherton, 2015), and the dynamics of stratocumulus clouds are particularly

hard to resolve (Schneider et al., 2019).

For further justification of our vortex identification algorithm, relevant vortex statistics are presented in Appendix B.

### 3.3 Climatology of Madeira vortex streets

In this section, we present the decadal analysis of vortex streets in the lee of Madeira Island. For the case study in Section 3.2, we required the center of vortices shedded from Madeira Island to be within the black dashed polygon in Fig. 15, to incorporate

as many vortices as possible. For the decadal analysis, we chose a smaller domain, shown by the red polygon in Fig. 15, in order to reduce the number of vortices shedded from the La Palma island in the Canary Archipelago. The vortex count shown





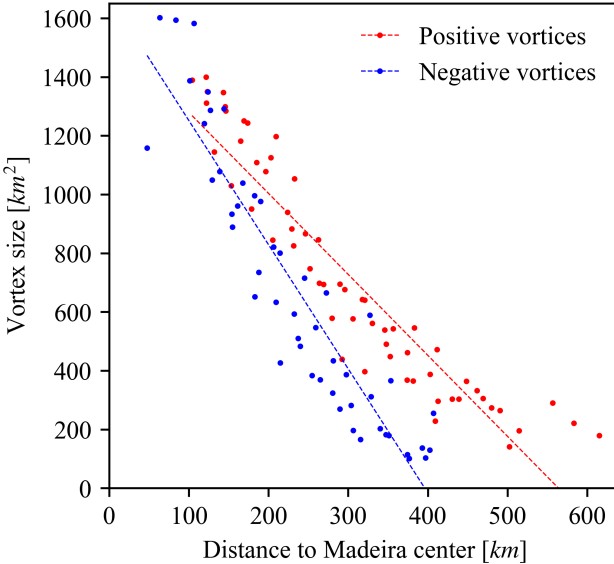

**Figure 11.** The size of four positive and four negative tracked vortices (see Supporting Information V4, Gao et al., 2021) against their distance to Madeira center. Two dashed lines are fitted to the scatter plots to show different decaying rates of positive and negative vortices.

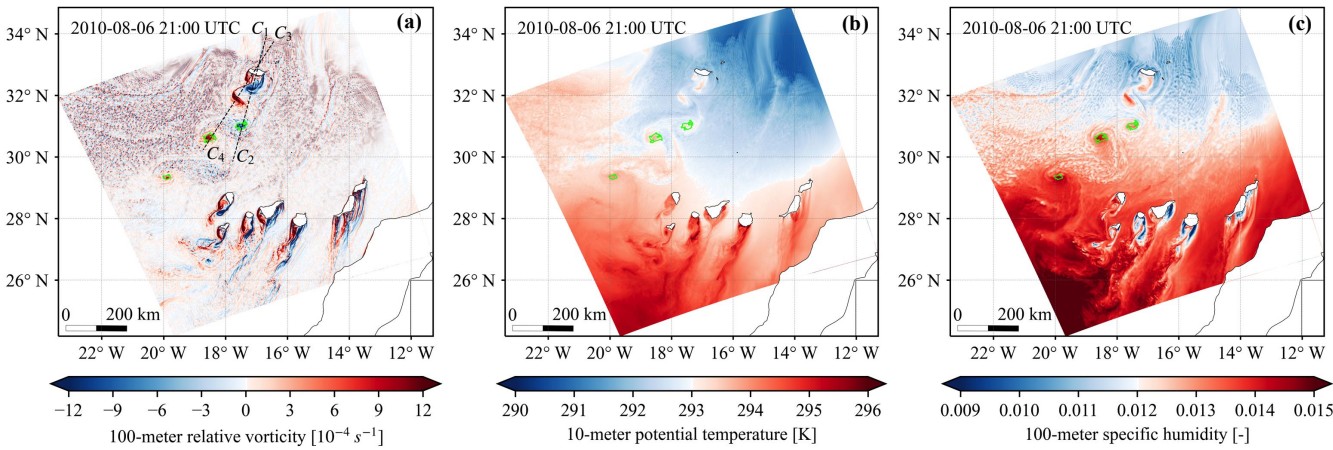

**Figure 12.** (a) 100-meter relative vorticity, (b) 10-meter potential temperature, and (c) 100-meter specific humidity at 2010-08-06 21:00 UTC. The vertical structures of the vortices were investigated along two cross-sections ($C_1$-$C_2$ and $C_3$-$C_4$) shown in (a). The identified vortices are encompassed by lime contours. The geographic map is from the Natural Earth.

in Fig. 15 shows a strongly increased frequency to the southwest of Madeira island, which corresponds to the lee during trade wind conditions.

 Figure 16a shows the daily vortex count during 2010 and the 10-day moving average. As the parameters of our algorithm were tuned based on data from February and August 2010, we validated it using data from November 2010 against manual





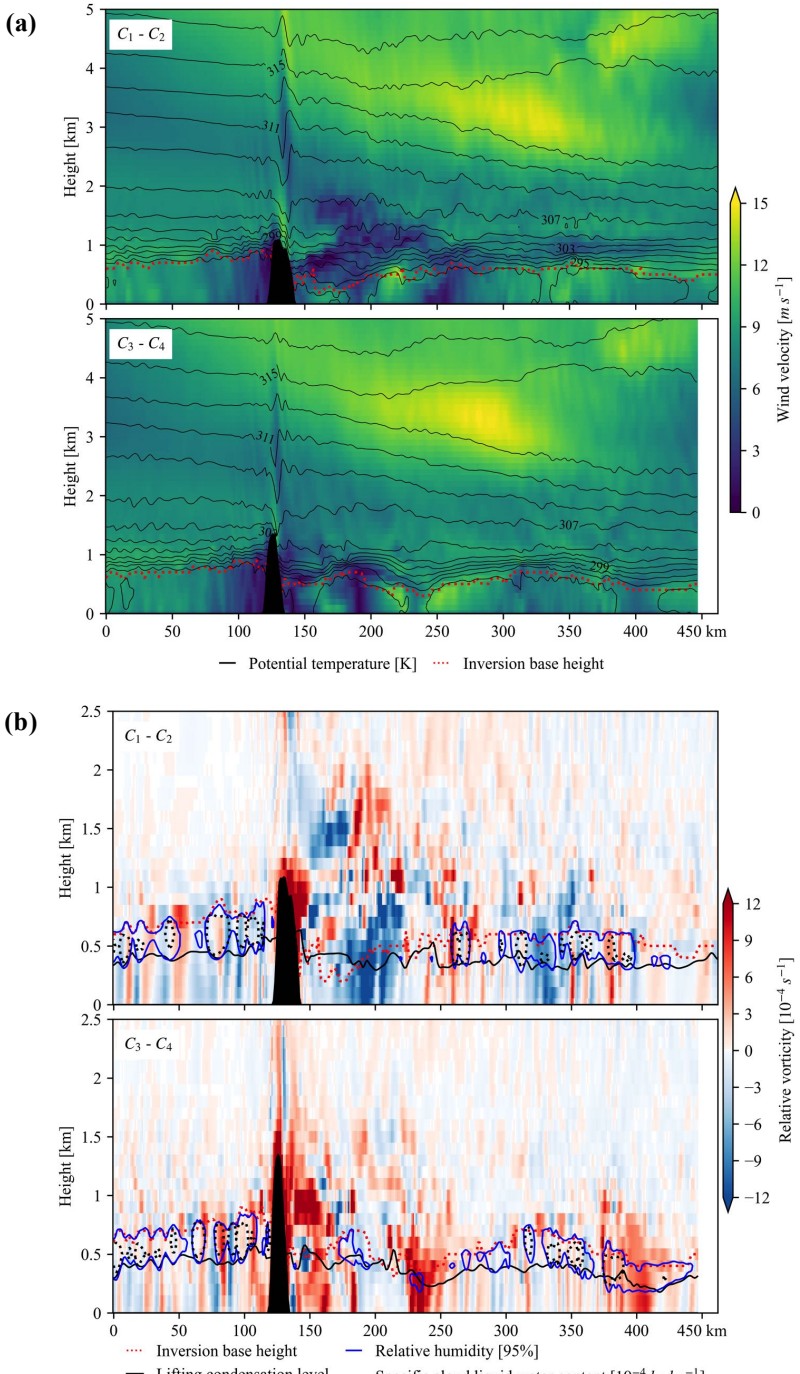

**Figure 13.** Vertical structure of (a) wind velocity, potential temperature, inversion base height, and (b) relative vorticity, relative humidity, lifting condensation level, and specific cloud liquid water content along two cross-sections $C_1$-$C_2$ and $C_3$-$C_4$ shown in Fig. 12a at 2010-08-06 21:00 UTC.





**Figure 14.** Comparison of satellite data (top row) versus simulation data (bottom row): (a) 10-meter winds and (b) the derived relative vorticity at a horizontal resolution of 6.25 km measured by ASCAT onboard Metop-A from EUMETSAT. The magenta polygons indicate the area with available ASCAT data over the analysis region. (c) Satellite image constructed with data observed by MSG Meteosat-9 from EUMETSAT. The NIR1.6, VIS0.8, and VIS0.6 channel data were used to represent red, green, and blue. The second row shows (d) 10-meter winds, (e) the derived relative vorticity, and (f) total cloud cover in CRS1 at a time point (2010-08-05 11:00 UTC) close to that of observations. The geographic map is from the Natural Earth.



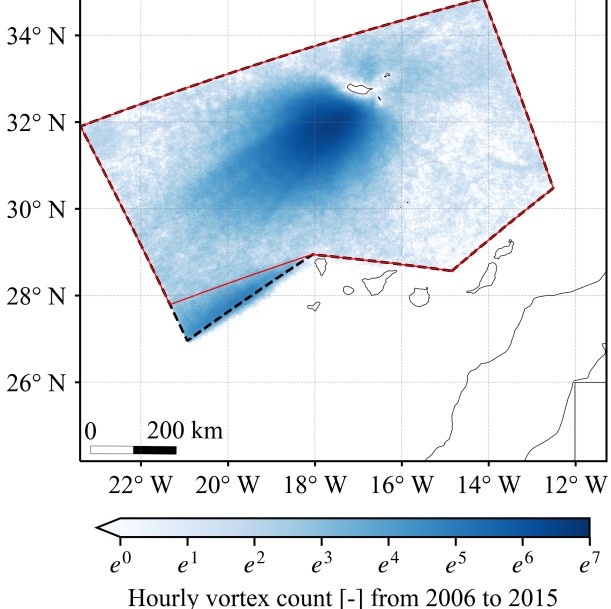

**Figure 15.** Hourly vortex count at each grid cell in CRS1 from 2006 to 2015 (totally 87,648 hours, $e^7 \approx 1097$). It shows the results when we used the dashed black polygon to restrict the region where the center of vortices shedded from Madeira Island can appear as described in Section 2.4. For each grid cell at each hour, we counted the vortex number as one if an identified vortex covers this grid cell. The exponential scale was used to account for the unevenly distributed frequency in vortices. The geographic map is from the Natural Earth.

analysis of the simulated vorticity field. During this month, there are 199 identified vortices and 171 correctly identified vortices, which correspond to a false positive rate (FPR) of 14% and one falsely identified vortex every 26 hours. Although this FPR does not reveal as good performance as 0.9% in the case study in Section 3.2, the second index 1/(26 h) still provides confidence in our algorithm. Figure 16b shows the monthly vortex count from 2006 to 2015, and the corresponding boxplots are

shown in Fig. 17, where several interesting phenomena can be observed. While Fig. 17a shows distinct interannual variations, Fig. 17b indicates a large monthly discrepancy. An increasing vortex shedding rate can be observed from April to August in Fig. 17b, as the wind direction is increasingly concentrated and the mean wind speed is increased Fig. 6 and the inversion height decreases from around 1.5 to 1 km in Fig. 7a. There are large variations and occasionally strong vortex shedding signals in March, which might result from abnormally lower inversion base in Fig. 7a and less precipitation in Fig. 5. Furthermore, we

can see a sharp transition from August to September resulting from sudden deceleration of winds in Fig. 6 and the increase of inversion height in Fig. 7a. This transition is also found in the study of Grubišić et al. (2015) by manually checking satellite images of eight years.




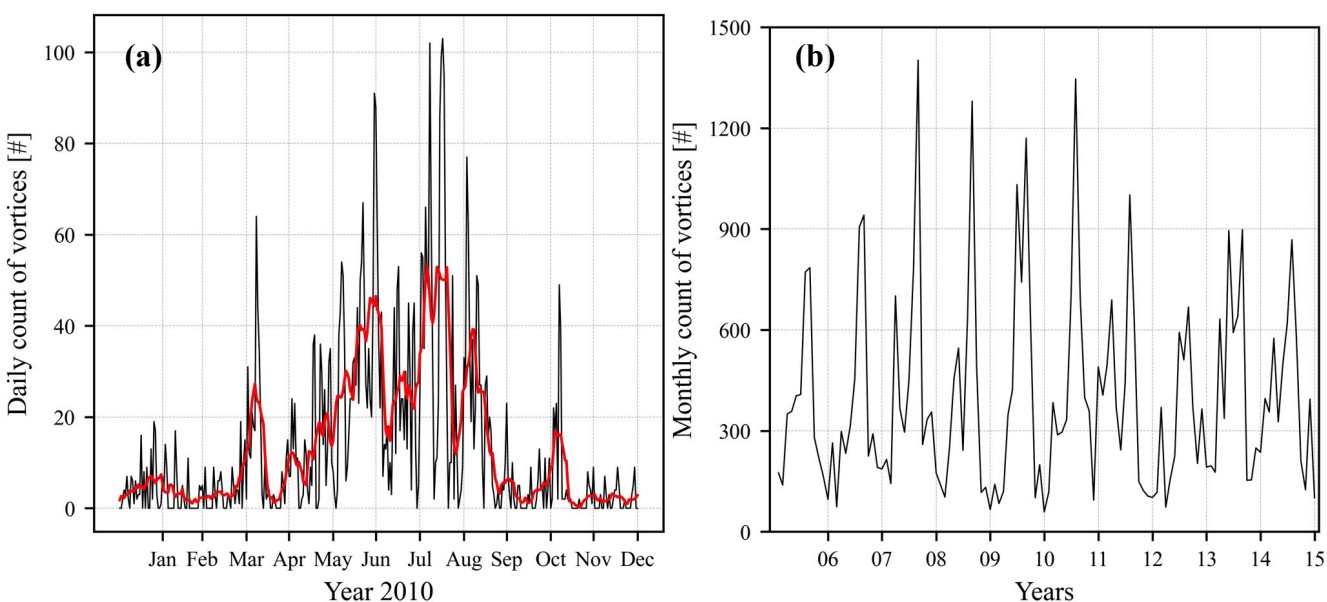

**Figure 16.** (a) Daily count of identified vortices in CRS1 in 2010 in black and its 10-day moving average in red. (b) Monthly count of identified vortices in CRS1 from 2006 to 2015.

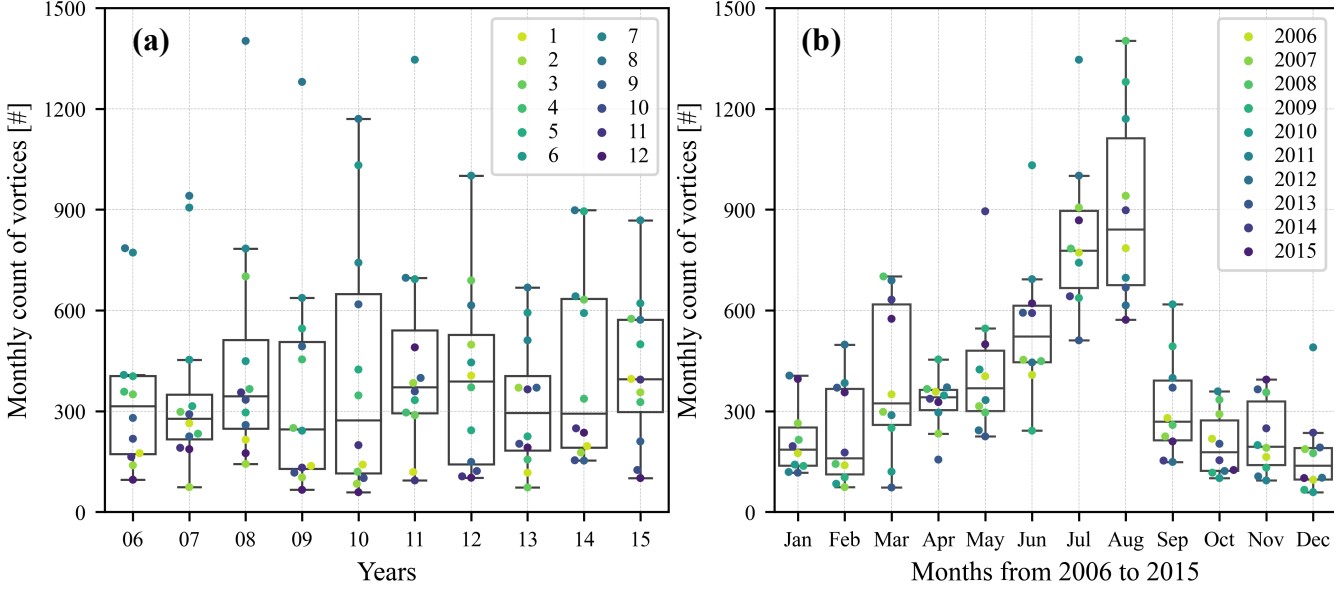

**Figure 17.** Boxplot of the monthly count of identified vortices in CRS1 from 2006 to 2015 in (a) each year and (b) each month. The boxes show quartiles of the data, and the whiskers extend to values within 1.5 times of the inter-quartile range from the low or high quartiles.



## 4    Conclusions

This study investigated vortex streets in the lee of Madeira Island during 2006-2015. We used the regional climate model
COSMO with a horizontal resolution of 1.1 km, driven by the ERA-Interim reanalysis. Firstly, we explored meteorological
conditions over our study area using the ERA5 reanalysis, model simulations, and radiosonde observations. Then we studied
basic properties of vortex streets based on a case study during 03-09 August 2010 and developed a vortex identification
algorithm. This algorithm was subsequently used for the climatological analysis of vortex streets over a 10-year period.

Although there are some discrepancies between simulated and observed precipitation, the added value of high-resolution
simulations that partly resolves the complex topography and convective processes is rather prominent. In particular, the CRS1
simulation shows more realistic spatial patterns and less negative biases over Madeira Island than the CPS12 simulation with
12 km horizontal resolution. The monthly wind conditions over our analysis region can be well simulated and reveal some
interesting phenomena. From April to August, the trade wind speed increases and the direction becomes better defined. Between
August and September the trades become much weaker again. In addition, the monthly patterns of inversion height obtained
from radiosonde observations at Funchal are also well captured in CRS1. From April to August the inversion height decreases
from 1.5 to 1 km, whereas it increases abruptly to above 1.6 km in September. Such monthly patterns of winds and inversion
height can be explained by the migration of the Hadley cell (Carrillo et al., 2016) and associated shifts of the Azores high.

A vortex shedding event behind Madeira Island during 03-09 August 2010 observed by Meteosat-9 of EUMETSAT is well
captured by CRS1. This event was chosen as a case study to investigate and validate basic properties of vortex streets in
realistic simulations. During this period, both synoptic and local atmospheric conditions were found to be conducive for vortex
shedding. The synoptic flow conditions from the ERA5 reanalysis reveal a direct dependence of vortex shedding in the lee of
Madeira Island on the location, shape, and magnitude of the Azores high. The upstream undisturbed 100-meter winds have a
velocity over $10 \ \mathrm{m \, s^{-1}}$ and show small spatial and temporal variations. In addition, both the dimensionless mountain height
and the dividing streamline height satisfy widely studied requirements for flow splitting and vortex shedding. Overall, the
numerical model indicates good performance in realistically simulating vortex shedding in this region and reveals interesting
properties of vortex streets. Firstly, the ratio of advection velocity of vortices to undisturbed wind velocity over $e_3$ was estimated
to be around 0.82 and the Strouhal number varies between 0.142 and 0.166, both values lying well within reported ranges in
literature. It is found that the negative vortices show faster decay rates than positive vortices. A potentially attractive explanation
is related to the role of the Coriolis force, which sharpens positive (cyclonic) vortices and widens negative (anticyclonic) ones.
The investigation of structures of vortex streets in realistic atmospheric simulations brought us several new insights. Although
positive potential temperature and specific humidity anomalies around vortices are well known for many vortex streets, the
fact that they can propagate further downstream than the dynamical (vorticity) contribution of the vortices has not yet been
widely noticed. Finally, the lowering of the capping inversion below the LCL is responsible for cloud-free regions in the lee of
Madeira Island.
Based on the case study, we developed a rule-based vortex identification algorithm. The algorithm was manually validated
and shows a good performance with few wrong identifications. This algorithm enabled us to draw a 10-year climatology of



vortex shedding to the lee of Madeira. Monthly patterns of the number of identified vortices shedded from Madeira Island are consistent with that of winds over our analysis region and inversion height observed at Funchal. These patterns are mainly governed by the annual cycle of the trades, Hadley circulation, and the Azores high. From April to August, the vortex shedding rate is increasing. This is driven by a strengthening of the trade winds and a lowering of the inversion height. Thereafter, the sudden weakening of the trade winds and an increase in inversion height in September lead to a pronounced and sharp drop in vortex shedding frequency, confirming the results of Grubišić et al. (2015) obtained from satellite images for eight years.

Lastly, we point out some limitations in our work and make several suggestions for future studies. Although the islands in the Canary Archipelago, especially La Palma and El Hierro, also show very nice vortex shedding signals, we do not consider multiple islands in one paper for simplicity and feasibility. While we used the six-hourly ERA-Interim reanalysis as initial and boundary conditions for CPS12, the employment of the ERA5 reanalysis with higher spatial and temporal resolution might improve the simulation of vortex streets. Although our work involved only 2D vortex identification, our algorithms can be extended for 3D vortex identification. The 3D vortex visualization, automatic vortex tracking, and vortex identification using deep learning could also be considered in further works. Finally, we noted some considerable challenges such as limitations in the simulation of the cloud cover in particular in areas of stratocumulus. Nevertheless, we believe that our model-based generation of a vortex-shedding climatology is overall succesful and of considerable interest.

*Code and data availability.* The ERA5 reanalysis can be obtained from the Climate Data Store (https://cds.climate.copernicus.eu). The ASCAT and MSG data are provided by EUMETSAT (https://navigator.eumetsat.int), and the ASCAT Wind Data Processor (AWDP) is available from the OSI SAF (https://nwp-saf.eumetsat.int). The radiosonde observations are from IGRA 2 (https://www.ncei.noaa.gov). The observational station data are provided by Portuguese Institute of the Sea and the Atmosphere and by Portuguese Environment Agency and it is available upon request. The MODIS satellite data are from the NASA website (https://ladsweb.modaps.eosdis.nasa.gov). The digital elevation model of Madeira Archipelago can be obtained from the European Environment Agency (https://sdi.eea.europa.eu). The maps in the figures are from the Natural Earth (https://www.naturalearthdata.com/) and the GADM (https://gadm.org/). The model simulations are available from the authors upon request. The code for analysis can be found in this GitHub repository: https://github.com/l975421700/DEoAI_analysis.

*Video supplement.* The Supporting Information (four videos) can be found at *Supporting Information: Vortex streets to the lee of Madeira in a km-resolution regional climate model* (https://doi.org/10.5281/zenodo.4462118).

## Appendix A: The wavelet analysis for denoising

Abramovich et al. (2000) focused on statistical applications of the wavelet analysis and could serve as a good start point for statisticians. All wavelets in the basis are derived from two mutually orthonormal parent wavelets through translations of the scaling function $\phi$ and dilations and translations of the mother wavelet $\psi$ as





$$\boldsymbol{\phi}_{j_0 k}(t) = 2^{j_0/2}\phi(2^{j_0}t - k),\tag{A1}$$

$$\boldsymbol{\psi}_{jk}(t) = 2^{j/2}\boldsymbol{\psi}(2^j t - k),\tag{A2}$$

where $j_0 \in Z$ is fixed, $Z$ is the set of integers, $k \in Z$, and $j = j_0, j_0 + 1, j_0 + 2, ...$

Intuitively we can regard $\phi$ as a kernel function and $\psi$ as a localized oscillation, where a unit increase of $j$ in Eq. (A2) doubles the frequency or resolution of $\boldsymbol{\psi}_{jk}(t)$. A unit increase of $k$ shifts the location of $\boldsymbol{\phi}_{j_0 k}(t)$ by $2^{-j_0}$ and $\boldsymbol{\psi}_{jk}(t)$ by $2^{-j}$.

Given a wavelet basis and assuming a signal to be infinite, $g(t)$ can be represented through a wavelet series as

$$\boldsymbol{g}(t) = \sum_{k \in Z} c_{j_0 k}\boldsymbol{\phi}_{j_0 k}(t) + \sum_{j=j_0}^{\infty}\sum_{k \in Z} w_{jk}\boldsymbol{\psi}_{jk}(t),\tag{A3}$$

where $c_{j_0 k} = \int_R \boldsymbol{g}(t)\boldsymbol{\phi}_{j_0 k}(t)dt$ and $w_{jk} = \int_R \boldsymbol{g}(t)\boldsymbol{\psi}_{jk}(t)dt$.

Considering discretely sampled signals $\boldsymbol{g} = (g(t_1), g(t_2), ..., g(t_n))^T$ (assume $n = 2^J$ where $J$ is a positive integer) at a constant interval, the DWT needs to be considered as

$$\boldsymbol{d} = \mathbf{W}\boldsymbol{g},\tag{A4}$$

where $\mathbf{W}$ is an $n \times n$ orthogonal matrix associated with selected wavelets, and $\boldsymbol{d}$ is an $n \times 1$ vector comprising discrete scaling coefficients $u_{j_0 k}$ ($\approx c_{j_0 k} \times \sqrt{n}$, $k = 0, 1, ..., 2^{j_0} - 1$) and discrete wavelet coefficients $d_{jk}$ ($\approx w_{jk} \times \sqrt{n}$, $j = j_0, j_0 + 1, ..., J - 1$, 495 $k = 0, 1, ..., 2^j - 1$).

The inverse DWT can be obtained based on the orthogonality of $\mathbf{W}$ as $\boldsymbol{g} = \mathbf{W}^T \boldsymbol{d}$ and used to denoise through reserving only a part of largest values in $\boldsymbol{d}$ (Wickerhauser, 1996; Ruppert-Felsot et al., 2005), where coefficients above (below) a threshold are assumed to correspond to signals (noises). The choice of the threshold depends on a measure of the number of significant coefficients $N_0$ defined as

$$N_0(\boldsymbol{g}) = e^{H(\boldsymbol{g})},\tag{A5}$$

where $H(\boldsymbol{g})$ is the entropy of $\boldsymbol{g}$ and defined as

$$H(\boldsymbol{g}) = -\sum_{i=1}^{n} \boldsymbol{p}_i \ln \boldsymbol{p}_i,\tag{A6}$$

where $\boldsymbol{p}_i = |\boldsymbol{g}_i|^2/||\boldsymbol{g}||^2$ and $||\boldsymbol{g}||^2 = \sum_i |\boldsymbol{g}_i|^2$.

As for the selection of parent wavelets, the function support (compact or infinite), orthogonality, smoothness (i.e. regularity
or the number of continuous derivatives), and the number of vanishing moments should be considered. Wang et al. (2018) gives
a good visualized introduction to wavelet families, while some recommendations and requirements about their selection can
be found in Domingues et al. (2005) and Jawerth and Sweldens (1994). The Haar basis is the oldest and simplest wavelet basis
for *g(t)*. Albeit the Haar basis does not have a good time-frequency localization, and the resulting wavelets are discontinuous
and thus unsuitable for smooth functions, its performance was equivalent to other wavelets in our preliminary analysis, as also
reported by Siegel and Weiss (1997). Therefore, it is selected for our further analysis for simplicity. In real cases, signals are
normally zero outside a finite interval. As simply setting the signals to be zero outside the finite interval results in discontinuities
at boundaries, periodic boundary handling is adopted as a solution.

## Appendix B: Identified vortex statistics

In Fig. B1, we displayed several properties of extracted vortices using the first two steps of our vortex identification algorithm,
identified vortices through subsequent criteria, and three falsely identified vortices during the case study period. Figure B1a
shows that the mean relative vorticity of identified vortices distributes well above the threshold $3 \times 10^{-4}$ s$^{-1}$ selected to identify
candidate grids. Some extracted vortices have mean relative vorticity below the selected threshold. Two reasons can explain
this phenomenon: These vortices have a mean size of 4 km$^2$ and thus are vulnerable to noises; The wavelet transform could
increase the relative vorticity above the threshold, whereas we extract the vortex characteristics based on the original data. We
can see a sharp edge of the size of identified vortices at the threshold of 100 km$^2$, which indicates some falsely rejected small
vortices. However, we expect those small vortices to have less relevance and accept the threshold to eliminate a large number
of noisy structures. Figure B1c-e reveals significant discrepancies between statistics of identified vortices and the thresholds
in our algorithm. This is a great signal that indicates our algorithm consists of loose criteria and does not implicitly determine
these vortex characteristics.

*Author contributions.* CS and JVT initiated and led the study. JVT conducted the decadal model simulation. QG and CZ contributed ideas
in the early stage of the study. QG performed the data analysis and created all figures and tables with contributions from CZ and DL. QG
wrote the manuscript with contributions from all authors.

*Competing interests.* The authors declare that they have no conflict of interest.

*Acknowledgements.* We acknowledge PRACE for awarding us access to Piz Daint at the Swiss National Supercomputing Centre (CSCS).
We also acknowledge the Federal Office for Meteorology and Climatology MeteoSwiss, CSCS, and ETH Zurich for their contributions to the
development of the GPU-accelerated version of COSMO. Daniela C.A. Lima would like to acknowledge the financial support by Pre-defined

 

**Figure B1.** Histograms of vortex statistics including (a) mean relative vorticity, (b) vortex size, (c) peak or maximum relative vorticity, (d) the ratio of the largest distance within a vortex to its nominal radius, and (e) deviation angles described in Section 2.4. The blue shadings represent extracted vortices using the first two steps of our algorithm, and the red (black) shadings represent the correctly (falsely) identified vortices using our algorithm during the case study period.

Project-2 National Roadmap for Adaptation XXI (PDP-2) funded by EEA-Financial Mechanism 2014-2021 and the Portuguese Environment Agency.



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
