# Peer review of "Vortex streets to the lee of Madeira in a km-resolution regional climate model"

_EGUsphere, 2022_

## Referee Comment (RC1)

**Comments** on the manuscript egussphere-2022-965 „Vortex streets in the lee of Madeira in a km-resolution regional climate model" by Gao et al.

**General comments**

The paper deals with atmospheric vortex streets in the lee of the island Madeira. The phenonemon of vortex streets has obtained considerable interest in the last decades, since vortex streets have been observed by the first meteorological satellites. The studies on vortex streets have been of the types pure observational using satellite images, purely theoretical on the physical mechanism of vortex shedding, numerical simulations of idealized situations concerning obstacle forms and basic flow conditions, numerical simulations for concrete observed cases for real islands and climatological studies concerning frequency of vortex streets appearance. Various examples are listed in the quite comprehensive reference list of this paper. Hence one might wonder, which new aspect on this topic the authors can present to the readership.

To the knowledge of the reviewer it is the first time, that a climatological statistic of vortex shedding for a concrete island (here: Madeira) has been established based purely on numerical simulations. In order to manage this, the authors had to develop an automatic vortex identification scheme for the output of a high resolution climate model. As this identification scheme is quite universal it can be applied also to other models or for the same model concerning other islands where vortex streets are quite common. The choice of Madeira as first object for a model-based vortex street climatology is well founded, as many studies of vortex streets for Madeira have been published and also a vortex climatology is available based on satellite observations (Grubisic et al, 2015).

In summary, the paper adds quite new information on the problem of atmospheric vortex shedding by islands and the methods developed herein can be applied to other vortex climatologies from numerical model outputs. Hence the paper is well recommended for publication subject to some minor changes as listed below.

**Special comments**

1.Vortex counting

The climatology of vortex shedding behind Madeira is obtained by hourly counting the vortices in the area shown in Fig. 15 applying the rules described in section 2.4. Now there are also daily counts, monthly counts etc. (see Figures 8, 16, 17). How are these obtained? Is the daily count the sum of the hourly counts etc.? The reason for this question is the following: Once a vortex is shed from the island, it will be counted every hour during its lifetime. During this time, it will move downstream and depending on its lifetime, the final position will vary in the counting area (Figure 15). The lifetime of all vortices during the 10 year period is reflected indirectly by the distribution of counted vortices shown in Figure 15. Now let us suppose, during one day only 1 vortex is shed from Madeira. This would appear in the hourly count as 1. In 24 hours it would be counted 24 times. Hence the daily count would

be 24, but there is only this one vortex in the area. So the daily count would not give the number of different vortices identified during one day but the number of the hourly positions of all vortices during this day. The same could hold for weekly and monthly or total count(?).

The authors are asked for clarifying their "counting method".

2.Temperature profiles

From the current study (as well from other studies) it is apperant, that the location of the temperature inversion in the marine atmospheric boundary layer (MBL) plays a crucial role in the formation of vortex streets behind islands (i.e. the inversion height must be less than the island height). It would therefore of advantage, if a typical vertical profile of potential temperature (and also wind speed) in the MBL for Madeira would be presented as derived from observations and from the model output. This should be done for day 2010-08-06 of the case study, for which many Figures are shown. By this one could also judge the performance of the model concerning strong inversions in the MBL.

2.Froude number-dimensionless mountain height

It is apparent from the literature, that the frequency of vortex shedding from islands is not only dependent in the height of the temperature inversion but also on the upstream wind speed. The combined effect is usually characterized by a Froude-number Fr or the dimensionless mountain height $h_{dim} = 1/Fr$ (see section 2.3). Vortex shedding is favourable for say e.g. $Fr \leq 0.3$ or $d_{dim} \geq 3$ or so. The authors show this for the case study in Figure 8, but it would be also of interest, if the relation is also valid for the whole climatology from the long term model runs. Hence if the data on Fr or $h_{dim}$ have been stored for the whole period, they should be presented e.g. in Figure 7, where the climatology for the inversion height is also shown. One could compare then $h_{dim}$ or Fr with the vortex counts in Figure 17b.

3.Inversion height and vortex shedding

From comparing Figure 7 with Figure 17b it seems apparent, that a low inversion height is favourable for vortex formation, whereas a high inversion is not.  But in the discussion of the case study (shown in Figure 8) on page 14 the authors write:

„In addition, a sudden increase of the inversion base over e3 in the solid blue line at 2010-08-06 05:00 UTC also precedes the increasing vortex shedding signal".

 Is this not a contradiction to Figs 7 and 17b? Or is the decrease of the wind speed during this period the reason for the increasing vortex signal?

---

## Author Response (AR1)

**Response to referee 1**

We thank the reviewer for reviewing the paper and the good comments and suggestions. We have addressed the specific comments below.

**Comment 1**:

Vortex counting.

The climatology of vortex shedding behind Madeira is obtained by hourly counting the vortices in the area shown in Fig. 15 applying the rules described in section 2.4. Now there are also daily counts, monthly counts etc. (see Figures 8, 16, 17). How are these obtained? Is the daily count the sum of the hourly counts etc.? The reason for this question is the following: Once a vortex is shed from the island, it will be counted every hour during its lifetime. During this time, it will move downstream and depending on its lifetime, the final position will vary in the counting area (Figure 15). The lifetime of all vortices during the 10 year period is reflected indirectly by the distribution of counted vortices shown in Figure 15. Now let us suppose, during one day only 1 vortex is shed from Madeira. This would appear in the hourly count as 1. In 24 hours it would be counted 24 times. Hence the daily count would be 24, but there is only this one vortex in the area. So the daily count would not give the number of different vortices identified during one day but the number of the hourly positions of all vortices during this day. The same could hold for weekly and monthly or total count(?).

The authors are asked for clarifying their "counting method".

**Reply 1**:

Thank you for this great question. We added description for counting methods in Sec. 2.4:

" *The quantification and climatology of vortex shedding behind Madeira relies on vortex counting. We used two kinds of counting methods serving different purposes. After vortices shed from Madeira are identified at each hour, the hourly number of vortices over the study domain can be obtained directly by counting different vortices (e.g. Fig. 7). Thereafter, the daily and monthly number of vortices are aggregated from the hourly numbers (Fig. 15 and Fig. 16). The second approach is applied to investigate the spatial distribution of vortex shedding (i.e. Fig. 14). For each grid cell at each hour, we counted the vortex number as one if any identified vortex covers this grid cell. Then the hourly vortex count at each grid cell is obtained by summing the vortex number through the whole study period (2006-2015). Consequently, both counting methods will result in the same vortex being counted multiple times, as the lifetime of a vortex typically is longer than one hour. In order to count unique vortices, Lagrangian tracking would have to be used already during the model simulation. However, this study is based on hourly model output. Therefore, an approach where we accept multiple counting is likely more robust than an approach with additional criteria for the uniqueness of vortices and will not qualitatively change the results of this study.*"

We agree that these methods will count the same vortex several times during its lifetime. It would also be interesting to know how many "different" vortices there are at daily and monthly scales. But for our study, we think the numbers we get are also interesting, as the lifetime of vortices also depends on synoptic scale conditions.

**Comment 2**:

Temperature profiles

From the current study (as well from other studies) it is apparent, that the location of the temperature inversion in the marine atmospheric boundary layer (MBL) plays a crucial role in the formation of vortex streets behind islands (i.e. the inversion height must be less than the island height). It would therefore of advantage, if a typical vertical profile of potential temperature (and also wind speed) in the MBL for Madeira would be presented as derived from observations and from the model output. This should be done for day 2010-08-06 of the case study, for which many Figures are shown. By this one could also judge the performance of the model concerning strong inversions in the MBL.

**Reply 2**:

We thank the reviewer for this comment. We add Fig. C2 to show vertical temperature profiles at Funchal from the radiosonde observations and the simulation at 2010-08-06 12:00 UTC. As the inversion height is calculated based on temperature profiles, we show temperature rather than potential temperature. Preliminary analysis on the vertical temperature profiles over 2006-2015 comparing observations and simulations indicate that this timestamp is typical of situations at Funchal. We also added description in Sec. 3.1: "*Typical vertical temperature profiles from the radiosonde observations and the simulation are given in Fig. C2. The simulated and observed inversion height in this example agrees relatively well but shows the same bias as in Fig. 6a.*"

Vertical structure of simulated wind speed can be found in Fig. 12. Radiosonde observations also provide wind speed data. As the wind at Funchal is significantly disturbed (Fig. 12), we did not compare observed and simulated wind speed at various vertical levels.

**Comment 3**:

Froude number-dimensionless mountain height

It is apparent from the literature, that the frequency of vortex shedding from islands is not only dependent in the height of the temperature inversion but also on the upstream wind speed. The combined effect is usually characterized by a Froude-number Fr or the dimensionless mountain height hdim = 1/Fr (see section 2.3). Vortex shedding is favourable for say e.g. Fr ≤ 0.3 or ddim ≥ 3 or so. The authors show this for the case study in Figure 8, but it would be also of interest, if the relation is also valid for the whole climatology from the long term model runs. Hence if the data on Fr or hdim have been stored for the whole period, they should be presented e.g. in Figure 7, where the climatology for the inversion height is also shown. One could compare then hdim or Fr with the vortex counts in Figure 17b.

**Reply 3**:

Thank you for the suggestion. We fully agree that it would be beneficial to have a decadal climatology of dimensionless mountain height, but we do not have the required data stored.

For the climatology of inversion height in Fig. 6, we used available data from the ellipse *se* in Fig. 1b, which is approximated as undisturbed upstream conditions. This is considered as an acceptable approximation as the inversion height does not vary significantly upstream of Madeira as shown in Fig. 12. But for dimensionless mountain height, we avoid such approximation to be more conservative.

**Comment 4**:

Inversion height and vortex shedding

From comparing Figure 7 with Figure 17b it seems apparent, that a low inversion height is favourable for vortex formation, whereas a high inversion is not. But in the discussion of the case study (shown in Figure 8) on page 14 the authors write:

„In addition, a sudden increase of the inversion base over e3 in the solid blue line at 2010-08- 06 05:00 UTC also precedes the increasing vortex shedding signal".

Is this not a contradiction to Figs 7 and 17b? Or is the decrease of the wind speed during this period the reason for the increasing vortex signal?

**Reply 4**:

Thank you. Yes, this sentence about Fig. 8 is confusing. We didn't mean a causality relationship but just wanted to point out the co-occurrence of two phenomena. To avoid the confusion, we deleted this sentence. The increasing vortex shedding signal is more likely related to the increase in dividing streamline height and the dimensionless mountain height.

**Response to referee 2**

We appreciate the efforts the referee put in reviewing this manuscript. Thank you very much for the detailed comments. We reply to each comment below.

**Comment 1**:

I might suggest omitting the details about the precipitation climatology. You could mention that you checked it, but I'd only focus on those parameters relevant to vortex shedding.

**Reply 1**:

Thank you, the evaluation of spatial patterns of annual mean precipitation is moved to appendix in Fig. C1. Figure 3 and 4 are reserved for two reasons: 1) Evaluate the performance of simulations at two horizontal resolution. As the topography is very important for the simulation of both precipitation and vortex shedding, it is important to know the improved performance with increased resolution. 2) The annual cycles of precipitation and vortex shedding share some common features, which might be influenced by the annual variations of the Azores high. Future works could consider to put these phenomena in a common framework for Madeira climate.

**Comment 2**:

Line 109: A grid spacing of 1.1 km is not convection resolving; rather, it is "convection allowing" (to resolve convective motions, a grid spacing of about 100 m is needed).

**Reply 2**:

We agree with the referee. Convection-resolving is changed to convection-allowing.

**Comment 3**:

Line 109, 121: In the modeling world, one distinguishes between grid spacing and resolution (the resolved scale is several times larger than the grid spacing). I saw you statement in line 143, but I would still recommend using "grid spacing" instead of "resolution "when talking about the model setup.

**Reply 3**:

Thank you. The term resolution related to model setup is changed to grid spacing.

**Comment 4**:

Throughout, replace "shedded" with "shed."

**Reply 4**:

Changed as suggested, thank you.

**Comment 5**:

Line 196-197: This needs to be justified better.

**Reply 5**:

Thanks a lot for this great point. We fully agree that it is relevant to justify when the dividing streamline concept is valid. However, such theoretical justification is out of the scope of the current study, so we modified the sentence to be more conservative:

*"Therefore, relationships between the dividing streamline height and vortex shedding should be investigated and interpreted with caution."*

**Comment 6**:

Line 222 ff. (vortex identification criteria): I found these criteria quite subjective; maybe you can use a typical vortex street to show, and justify, why these criteria were used? Did you also consider criteria such as the kinematical vorticity number or Okubo-Weiss number? If not, why not?

**Reply 6**:

Thank you for this question. We understand that these criteria sound subjective, but all of them are based on observations from the case study in Sec.3.2. The reasons why these criteria are used are justified partly right after the criteria, and partly though Fig. B1. Typical vortices can be found in Fig. 11. Subjectivity is one of the disadvantages related to this kind of vortex identification (i.e. using relative vorticity), albeit we tried to be objective while determining the criteria for our research purpose.

Kinematical vorticity number or Okubo-Weiss number relating deformation and rotation of the flow might be a way to be more objective and has been applied for the identification of extratropical cyclones (Lisa et al. 2016, https://doi.org/10.3402/tellusa.v68.29464 ). Currently, it is more of an alternative rather than complementary approach to vorticity-based vortex identification. We add one sentence in the Conclusion section to point out its potential for future work: "The kinematic vorticity number could also be considered for vortex identification in future work (Schielicke et al., 2016)".

**Comment 7**:

Section 3.2: You refer to Supporting Information, rather than showing images. It is good to have those animations, but the reader ought to be able to follow the presentation by what's provided in the paper. Pleaser include snapshots of these animations that allow the reader to follow along.

**Reply 7**:

We appreciate this comment. As we already have equivalent snapshots of the animations in Fig. 9 and 11., it might be a bit redundant to add additional snapshots and make the manuscript lengthy. So we add in the text: "Snapshots of this vortex shedding event can be found in Fig. 9 and Fig. 11.".

**Comment 8**:

In figure captions and in the text, please make it clear when you discuss modelled vortices or observed ones.

**Reply 8**:

Thank you for this suggestion. We add in the text: "When discussing vortex streets, we refer to modelled vortices by default;

otherwise, we stress that the vortices are based on observations."

**Comment 9**:

Line 19: Turbulent flows are well-mixed and hence neutrally stratified in the atmosphere, no?

**Reply 9**:

Thank you for this question. Yes, turbulent flows are well-mixed. Here stably-stratified refers to the stratification below and above the inversion layer.

**Comment 10**:

Line 30: Do you mean "lifetime of individual vortices"?

**Reply 10**:

Yes, changed as suggested, thank you.

**Comment 11**:

Line 38: This seems like a very specific result from that study—is this relevant background for readers to follow your study?

**Reply 11**:

Yes, we also estimated the ratio of vortex shedding speed and background flow speed (0.82±0.02). Such comparison reveals good agreement so it should be useful to mention it here. And it also gives a more comprehensive overview regarding research directions in this field.

**Comment 12**:

Line 42: Why is there convergence at the surface when there is subsidence above?

**Reply 12**:

Thank you for this question. There is divergence above. They mentioned a sinking inversion, which means a lower inversion rather than subsidence. We changed it to be unambiguous.

**Comment 13**:

Line 44: Don't understand what "turbulent stresses are tilted" means

**Reply 13**:

Thank you for this question. We actually meant turbulent stresses and the tilting of horizontal vorticity. We changed the order to be clearer: "From a mathematical perspective, the vertical vorticity can be derived from turbulent stresses and the tilting of horizontal vorticity (Epifanio, 2015; Rotunno et al., 1999)".

**Comment 14**:

Line 44: Rotunno didn't suggest that turbulent stresses is inessential, but that the vorticity is generated via baroclinity rather than surface drag (but turbulent stresses were still needed to tilt the vorticity if I understand correctly).

**Reply 14**:

Thank you for the correction. Yes, it is modified accordingly as shown in Reply 13.

**Comment 15**:

Line 71: Suggest "conserved" instead of "conservative" here.

**Reply 15**:

Thank you. Changed as suggested.

**Comment 16**:

Line 83: Suggest focus instead of focused.

**Reply 16**:

Thank you. It is changed.

**Comment 17**:

Line 94: Are these extreme events related to vortex shedding? Otherwise it is not clear why these events make Madeira a good location to study vortex streets.

**Reply 17**:

Thank you for this question. These two kinds of events result from north-easterlies through orographic lifting and flow-splitting, respectively. While there is no direct link, a better understanding of the land-air interactions should be beneficial for the studies of both phenomena.

**Comment 18**:

Line 98-99: I'll leave it up to you, but I'd let the readers decide whether the present study makes a "significant contribution."

**Reply 18**:

Thank you. I agree it is better to let the readers to decide, so I omit our judgment here.

**Comment 19**:

Line 110: Reword: ... without employing the shallow and deep convection schemes.

**Reply 19**:

Thank you. It is modified as suggested.

**Comment 20**:

Lin 196: Use "a" gravity wave aloft? Also, a gravity wave is associated with vertical displacements, no?

**Reply 20**:

Thank you. It is changed to "a" gravity wave. They meant deceleration is not related to vertical displacements of air parcels. It is modified to be clearer: "They argued that the deceleration of air parcels approaching the mountain is due to the pressure gradient associated with a hydrostatic gravity wave aloft, rather than due to vertical displacements of air parcels.".

**Comment 21**:

Line 109: Please define the "crosswise island diameter."

**Reply 21**:

Thank you for this comment. Crosswind island diameter means the width of the island perpendicular to the wind direction. It is added in the text: "where D is the width of the island perpendicular to the wind direction at the inversion base".

**Comment 22**:

Line 246: Please clarify the criterion, which to me reads like "< 450, 450-900, or > 900 km$^2$"

**Reply 22**:

Yes, it is modified: "If the vortex size is in the range of <450, 450-900, or >900 km$^2$, the angle between the mean wind vector within a vortex and the vector connecting the Madeira center and vortex center should be less than 30, 40, or 50°, respectively".

**Comment 23**:

Line 260 ff: Why are the precipitation amounts relevant in the context of vortex shedding? Just to gauge the quality of the simulations? Can the climatology be shortened? I'm not sure what purpose it serves.

**Reply 23**:

We appreciate this question. Precipitation is relevant on two aspects: 1) Yes, it is useful to evaluate the simulations in two different resolutions and reveal the improvements with a higher resolution. 2) As both precipitation and vortex shedding normally result from north-easterlies through orographic lifting and flow splitting, their annual cycles show some consistency. These might be controlled by the annual cycle in the location, strength, and shape of the Azores high. Future studies might put these phenomena in a common framework for Madeira climate.

**Comment 24**:

Line 196: We also show...

**Reply 24**:

Thank you, changed as suggested.

**Comment 25**:

Line 329: "within the ellipse" is ambiguous; is this along the major axis?

**Reply 25**:

We agree with the referee. It means averaged over the ellipse. It is modified in the manuscript, e.g. "averaged over the ellipese".

**Comment 26**:

Line 345: You mention that the isobars are "deformed toward a circle, which led to a stronger north-easterly flow." Are you suggesting that the symmetry is related to the wind speeds? Please explain (or delete the phrase).

**Reply 26**:

We thank the reviewer for this comment. We didn't mean that symmetry is related to the wind speeds. To avoid ambiguity it is deleted.

**Comment 27**:

Line 347: Vanishment -- > weakening

**Reply 27**:

Thank you, changed as suggested.

**Comment 28**:

Line 408: Reword: ... does not suggest as good a performance... or something like that.

**Reply 28**:

Thanks. It is modified as suggested.

**Comment 29**:

Line 400: Perhaps you mean you did not wish to include vortices shed from other islands?

**Reply 29**:

Yes, we rephrased it to be clearer: "For the decadal analysis, we chose a smaller domain, shown by the red polygon in Fig. 14, to exclude vortices shed from the La Palma island in the Canary Archipelago.".

**Comment 30**:

Line 438: Conducive to.

**Reply 30**:

Thank you. Changed as suggested.

**Comment 31**:

Line 444: widens -> weakens?

**Reply 31**:

Thank you, changed as suggested.

**Comment 32**:

Line 466: Again, my suggestion is to avoid statements such as "our results are of considerable interest" – time will tell.

**Reply 32**:

We agree with the reviewer. The suggestion is taken and we deleted such statements.

**Comment 33**:

Line 514: displayed -> display

**Reply 33**:

Thank you, changed.

**Comment 34**:

518: noises -> noise

**Reply 34**:

Thank you. Changed.

**Comment 35**:

Fig. 8, 2$^{nd}$ row: Is this the shedding **rate**? If it is the "number of vortices shed" it should be an ever increasing number.

**Reply 35**:

Thank you. It is the number of vortices detected over the study domain.

**Comment 36**:

Fig. 13: Why not use (shaded) contour plots instead of the pixel plot?

**Reply 36**:

Thank you for this comment. Contour plots post-process the gridded dataset to determine the contour positions. However, such post-processing does not justify itself as absolutely advantageous (although it might look better), so shaded contour plots might not be superior to pixel plot. We chose to display the original data without any post-processing.